# Variegated overexpression of chromosome 21 genes reveals molecular and immune subtypes of Down syndrome

Micah G. Donovan[1,5], Neetha P. Eduthan[1,5], Keith P. Smith[1], Eleanor C. Britton[1], Hannah R. Lyford [1], Paula Araya[1], Ross E. Granrath[1], Katherine A. Waugh[1,2], Belinda Enriquez Estrada[1], Angela L. Rachubinski [1,3], Kelly D. Sullivan [1,4], Matthew D. Galbraith [1,2] ✉ & Joaquin M. Espinosa [1,2] ✉

Individuals with Down syndrome, the genetic condition caused by trisomy 21, exhibit strong inter-individual variability in terms of developmental phenotypes and diagnosis of co-occurring conditions. The mechanisms underlying this variable developmental and clinical presentation await elucidation. We report an investigation of human chromosome 21 gene overexpression in hundreds of research participants with Down syndrome, which led to the identification of two major subsets of co-expressed genes. Using clustering analyses, we identified three main molecular subtypes of trisomy 21, based on differential overexpression patterns of chromosome 21 genes. We subsequently performed multiomics comparative analyses among subtypes using whole blood transcriptomes, plasma proteomes and metabolomes, and immune cell profiles. These efforts revealed strong heterogeneity in dysregulation of key pathophysiological processes across the three subtypes, underscored by differential multiomics signatures related to inflammation, immunity, cell growth and proliferation, and metabolism. We also observed distinct patterns of immune cell changes across subtypes. These findings provide insights into the molecular heterogeneity of trisomy 21 and lay the foundation for the development of personalized medicine approaches for the clinical management of Down syndrome.

Down syndrome (DS), the genetic condition caused by triplication of human chromosome 21 (HSA21), also known as trisomy 21 (T21), is characterized by highly variable developmental phenotypes, including stunted growth, organ dysmorphogenesis, neurodevelopmental delays, and cognitive differences[1]. Moreover, T21 confers an increased risk for a range of co-occurring medical conditions across the lifespan, including congenital heart defects, autism spectrum disorders, seizure disorders, hearing and vision problems, gastrointestinal issues, multiple autoimmune conditions, leukemias, and Alzheimer's disease, also with strong inter-individual variability[2–6]. Despite many advances in the clinical management of DS that have drastically extended life expectancy and improved quality of life in this population[7], the lack of mechanistic understanding about factors that influence the heterogeneous developmental and clinical impacts of T21 impede the development of personalized medicine approaches to further benefit people with DS.

[1]Linda Crnic Institute for Down Syndrome, University of Colorado Anschutz Medical Campus, Aurora, USA. [2]Department of Pharmacology, University of Colorado Anschutz Medical Campus, Aurora, USA. [3]Department of Pediatrics, Section of Developmental Pediatrics, University of Colorado Anschutz Medical Campus, Aurora, USA. [4]Department of Pediatrics, Section of Developmental Biology, University of Colorado Anschutz Medical Campus, Aurora, USA. [5]These authors contributed equally: Micah G. Donovan, Neetha P. Eduthan. ✉e-mail: matthew.galbraith@cuanschutz.edu; joaquin.espinosa@cuanschutz.edu

Establishing cause-effect relationships between overexpression of HSA21 genes and the spectrum of traits associated with DS has proven difficult. Key exceptions include the established connection between triplication of the amyloid precursor protein (*APP*) gene and the high prevalence of Alzheimer's disease in DS[8], and the well-defined contributions of interferon receptor genes[9] and the *DYRK1A* gene[10] to multiple phenotypes in mouse models of DS. Nevertheless, genes encoded on HSA21 are likely to have complex interactions, with potential for both cooperative and antagonistic relationships, which likely affect phenotypic outcomes for polygenic traits. Therefore, investigations into the contributions of different HSA21 genes, alone or in combination, to the phenotypic variability of DS are needed.

Within this context, we investigated the expression patterns of genes encoded on HSA21 in hundreds of individuals with T21 through whole blood transcriptome analysis. We observed significant variability regarding overexpression levels of HSA21 genes, leading to the discovery of two distinct HSA21 gene clusters. Using a consensus clustering algorithm, we identified three distinct molecular subtypes (MS) of DS based on unique expression patterns of HSA21 genes. To gain an understanding of the biological implications of these subtypes, we conducted comprehensive multiomics profiling using whole blood transcriptomics, plasma proteomics and metabolomics, and immune cell profiling via mass-cytometry. These efforts revealed that the molecular subtypes of DS display distinct multiomic landscapes, with clear differential dysregulation of key pathophysiological processes, metabolic pathways, and immune homeostasis. These findings provide the foundation for future investigations into the functional consequences of variegated HSA21 gene overexpression and the potential clinical implications of these findings toward personalized medicine approaches in DS.

## Results

### Individuals with Down syndrome show variegated overexpression of genes encoded on chromosome 21

In order to investigate inter-individual variability of HSA21 gene overexpression among individuals with DS, we analyzed an integrated multiomics dataset from the Human Trisome Project (HTP) cohort study derived from 356 research participants with DS and 146 age- and sex-matched euploid controls (D21) (Supplementary Fig. 1a). In a recently published analysis of the HTP whole blood RNA-seq dataset, which comprises transcriptomes from a subset of 304 individuals with DS and 96 euploid controls (D21), we identified ~10,000 differentially expressed mRNAs in the bloodstream of people with T21 encompassing nearly 2/3 of expressed genes[11]. Of the 126 protein-coding mRNAs and 42 lncRNAs encoded on HSA21[12] that were detected in this experiment, ~90% (151/168) displayed significant upregulation in individuals with DS, with an average fold-change of 1.51 (Supplementary Fig. 1b, c). These results are consistent with the expected effect of increased HSA21 gene dosage in T21. However, when plotting the expression distributions and ranking the 126 protein-coding genes by their expression relative to euploid controls, we observed pronounced inter-individual variability in HSA21 gene overexpression (Fig. 1a). Not only do individual genes show a wide range of average overexpression relative to euploid controls, but we also find that for every HSA21 gene, a subset of individuals displays expression close to or even below the mean levels observed in euploids (Fig. 1a). Furthermore, individuals with T21 showing very high or no overexpression are different for distinct genes. This is exemplified by participants HTP0484A and HTP0594A. Whereas HTP0484A has among the highest expression levels of *TSPEAR*, they display the lowest degree of expression for *COL6A2*, well below the range of euploid controls (Fig. 1b). In contrast, participant HTP0594A is the lowest expressor of *TSPEAR* but is among the top expressors of *GET1* (Fig. 1b).

These diverse profiles of HSA21 gene expression prompted us to analyze the co-expression patterns of HSA21 genes among individuals with DS in more detail. Toward this end, we generated a matrix of Spearman correlations using age- and sex-adjusted mRNA expression data for all 126 protein-coding genes encoded on HSA21 (Fig. 1c, Supplementary Data 1). Unsupervised hierarchical clustering of these correlations shows two main HSA21 gene clusters, a major cluster 1 composed of 97 genes and a minor cluster 2 of 29 genes (Fig. 1c). Within each cluster we observe robust co-expression patterns, with clear negative correlations between clusters (Fig. 1d). These relationships are demonstrated by numerous examples including strong positive correlations for *PAXBP1* (HSA21 gene cluster 1) versus *BRWD1* (HSA21 gene cluster 1) and *N6AMT1* (HSA21 gene cluster 1) (Fig. 1e); strong positive correlations for *IFNGR2* (HSA21 gene cluster 2) versus *IL10RB* (HSA21 gene cluster 2) and *KCNJ15* (HSA21 gene cluster 2) (Fig. 1f); and negative correlations for *PAXBP1* versus *IFNGR2* (Fig. 1g). Importantly, genes in HSA21 clusters 1 and 2 are dispersed throughout the chromosome, without any obvious spatial grouping (Supplementary Fig. 1d).

These results reveal that despite a common chromosomal residence and increased dosage in T21, there are distinct groups of HSA21 genes based on their expression pattern, indicative of diverse regulatory mechanisms governing these gene sets.

### Variable chromosome 21 gene expression distinguishes molecular subtypes in Down syndrome

Upon observing two main clusters of co-expressed HSA21 genes, we investigated whether differences in HSA21 gene expression patterns could identify distinct individuals in the T21 cohort. To that end, we performed consensus clustering, on the age-, sex- and source-adjusted mRNA expression values for the 126 protein-coding genes encoded on HSA21 across the 304 individuals with T21 (Supplementary Fig. 2a). This revealed that while a viable clustering solution is apparent at k = 2, the increase in the relative change in area under the curve is most pronounced at k = 3, suggesting that k > 3 does not significantly enhance the clarity or robustness of the clustering solution (Supplementary Fig. 2b, c). Henceforth these three groups of individuals with T21 are referred to as molecular subtypes (MS) 1-3. Interestingly, these subtypes are clearly distinguished based on their expression profiles of HSA21 genes (Fig. 2a, Supplementary Data 2). This is further demonstrated by separation of the subtypes in principal component analysis (PCA) of HSA21 gene expression (Fig. 2b). Notably, the subtypes do not show significant differences in age, sex or body mass index (BMI) (Fig. 2c–e, Supplementary Data 2), suggesting variability across MS may be driven by other mechanisms. We then calculated polygenic expression scores for HSA21 gene clusters 1 and 2 by summing the expression z-scores relative to euploids for the genes in each cluster. These polygenic scores demonstrated that MS1 exhibits the highest expression levels for HSA21 cluster 1 genes, followed by MS2 and then MS3 (Fig. 2f, g). This is exemplified by expression of the genes *ATP5PF*, *GABPA*, and *PAXBP1* across the three subtypes (Fig. 2h, top row). Conversely, whereas MS3 shows the highest expression for HSA21 cluster 2 genes, there is no significant difference between MS1 and MS2 (Fig. 2f), as demonstrated by expression of the interferon receptors *IFNAR1*, *IFNGR2*, and *IL10RB* (Fig. 2h, bottom row). At the gene level, individuals in MS2 show a hybrid pattern, with some HSA21 cluster 1 genes being expressed at levels similar to those observed for MS1 (e.g., *EVA1C*), some HSA21 cluster 2 genes similar to MS3 (e.g., *ITGB2*) and some genes with distinct overexpression (e.g., *PFKL*) (Fig. 2h, middle row). Importantly, while these genes are, on average, overexpressed across the entire T21 cohort (Supplementary Fig. 2d), their average expression across subtypes displays significant variability.

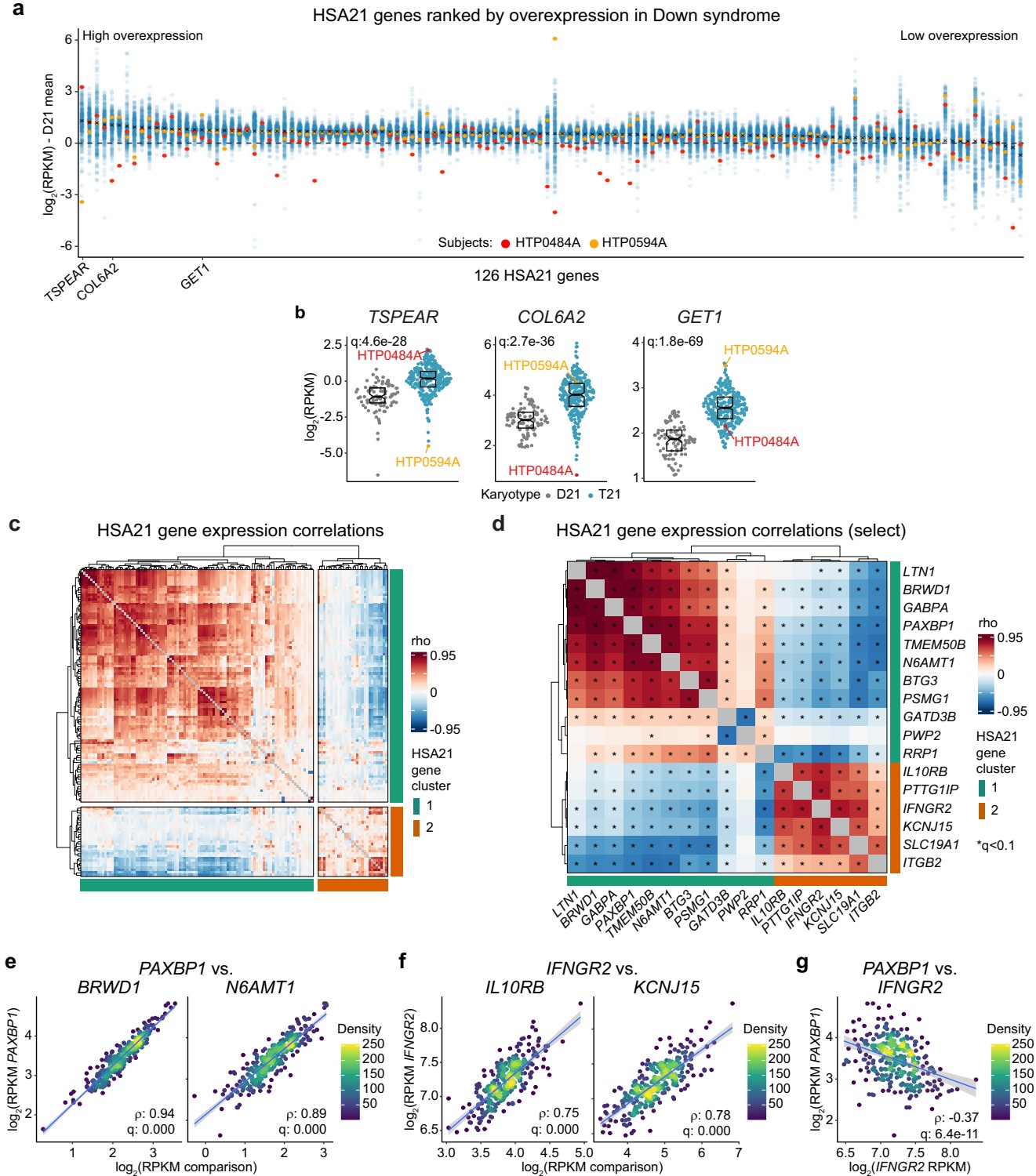

Collectively, these results demonstrate that although all individuals with DS carry an extra copy of HSA21, variegated overexpression of distinct HSA21 gene clusters leads to unique subgroups within this population.

### Molecular subtypes of Down syndrome are distinguished by distinct transcriptomic landscapes

Upon observing that distinct subtypes of individuals with T21 could be distinguished based on unique expression profiles of HSA21 genes, we undertook a comparative multiomics analysis using the HTP datasets. Toward this end, we analyzed by DEseq2, using adjustments for age and sex, differential expression of whole blood transcriptomes from each subtype relative to euploid controls and to each other (Supplementary Fig. 3a, Supplementary Data 2). These analyses revealed all three subtypes show widespread transcriptomic differences when compared to euploid controls, with thousands of both up- and down-regulated genes (Supplementary Fig. 3a). Interestingly, when comparing among subtypes, we also observed a similar magnitude of transcriptomic changes, whereby most transcriptome changes from the euploid baseline are unique to

**Fig. 1 | Individuals with Down syndrome show variegated overexpression of genes encoded on chromosome 21. a** Sina plots showing expression of chromosome 21 (HSA21) protein-coding genes (*n* = 126) in individuals with trisomy 21 (T21, *n* = 304) relative to euploid controls (D21, *n* = 96). Genes are ranked from left to right by decreasing degree of overexpression relative to D21, based on DESeq2 analysis. X represents mean expression for each individual gene. Dashed horizontal line indicates mean value in euploids. Two research subjects (HTP0484A, red; and HTP594A, yellow) are colored as indicated to illustrate differential expression patterns. **b** Sina plots showing expression of *TSPEAR*, *COL6A2* and *GET1* in D21 (*n* = 96, gray) and T21 (*n* = 304, teal). q-values are derived from DESeq2 comparisons to D21 and adjusted using the Benjamini−Hochberg method. Boxes represent interquartile ranges and medians, with notches approximating 95% confidence intervals. Red and yellow highlight distinct research participants. **c** Heatmap displaying unsupervised clustering of HSA21 gene expression Spearman correlations in individuals with DS (*n* = 304). Colors indicate HSA21 gene cluster 1 (teal) and cluster 2 (orange). **d** Heatmap highlighting the top 5 strongest positive and negative correlations from each HSA21 gene cluster in (**c**). Colors indicate HSA21 gene cluster 1 (teal) and cluster 2 (orange). Asterisks indicate significant Spearman correlations (q < 0.1) after Benjamini−Hochberg adjustment for multiple hypothesis testing. **e**−**g** Scatter plots showing in individuals with DS (*n* = 304) relationships between HSA21 cluster 1 genes: *PAXBP1* vs. *BRWD1* and *N6AMT1* (**e**), HSA21 cluster 2 genes: *IFNGR2* vs. *IL10RB* and *KCNJ15* (**f**), and *PAXBP1* vs. *IFNGR2* (**g**). *rho* and q-values (Benjamini−Hochberg adjusted *p*-values) for Spearman correlation are denoted. Points are colored by density; blue lines represent the fitted values from linear regressions, with 95% confidence intervals in grey.

each subtype (Fig. 3a, Supplementary Fig. 3a). In order to probe for differences in signaling pathways among these transcriptomic profiles, we utilized Ingenuity Pathway Analysis (IPA) of canonical pathways and upstream regulator networks on DEseq2 results comparing the full T21 cohort and individual subtypes to euploid controls (Fig. 3b, c, Supplementary Data 2). This exercise demonstrated that the predicted activation/inhibition of key biological pathways and master regulators of gene expression is variable in T21, which is underscored by marked differences across subtypes. When considering the full T21 cohort, we observe transcriptomic signatures indicative of heightened immune and inflammatory processes. This is demonstrated by predicted activation of phagosome formation, cytokine response to infections, and interferon signaling (Fig. 3b), concurrent with increased activity of key regulators of interferon-stimulated gene expression, namely NONO (non-POU domain-containing octamer-binding protein) and multiple IRFs (interferon regulatory factors) (Fig. 3c). IPA on the full T21 cohort also shows enrichment of signatures related to cellular metabolism and proliferation, including activation of the canonical oxidative phosphorylation pathway in addition to MYC and cyclin D1 upstream regulator networks (Fig. 3b, c). Interestingly, when considering each subtype, we observed that the signatures associated with T21 status are predominately driven by distinct subtypes. For example, whereas MS1 displays prominent enrichment of oxidative phosphorylation genes, as well as genes regulated downstream of MYC and cyclin D1, these signatures are not observed or are dampened in MS3 (Fig. 3b, c). MS1 also displays signatures indicative of enhanced EIF2 signaling and increased activity for MLXIPL (MLX-interacting protein 1), a member of the Myc/Mad/Max family of transcription factors, and ESRRA (Estrogen Related Receptor Alpha), which is not apparent when considering the entire T21 cohort (Fig. 3b, c). These pathway-level changes are demonstrated by stark differences across subtypes in the expression of genes involved in cellular proliferation (e.g., *NRAS*), translation (e.g., *EIF4E*), oxidative phosphorylation (e.g., *CYCS*), and ribosome assembly (e.g., *RPS27L*) (Fig. 3d, Supplementary Fig. 3b). In contrast, MS3 is marked by pronounced elevation of multiple immune- and inflammatory-related signatures, which are absent or dampened in MS1 (Fig. 3b, c). This is evident by the expression of genes associated with cytokine signaling (e.g., *CXCR1*), interferon responses (e.g., *IRF1*), the inflammasome complex (e.g., *NLRC4*), toll-like receptor signaling (e.g., *MYD88*), and the NF-κB pathway (e.g., *IKBKG*) (Fig. 3e, Supplementary Fig. 3c).

Considering the unique HSA21 gene expression profiles and global transcriptome signatures of each subtype, we analyzed correlations of HSA21 gene expression against the entirety of the transcriptome and subsequently analyzed the derived correlations via GSEA (Fig. 3f, Supplementary Fig. 3d). This analysis revealed that transcriptome signatures dysregulated in DS align with expression of genes in either HSA21 gene cluster 1 or HSA21 gene cluster 2. Moreover, gene signatures enriched in MS1, such as oxidative phosphorylation and MYC targets, are positively associated with expression of HSA21 cluster 1 genes, and negatively associated with expression of HSA21 cluster 2 genes (Fig. 3f). This pattern is evident for several non-HSA21 genes overexpressed in MS1, as illustrated by the relationship between expression of *NRAS* (encoded on chromosome 1) versus *PAXBP1* (HSA21 cluster 1), and *SLC19A1* (HSA21 cluster 2) (Fig. 3g, h, Supplementary Fig. 3e). In contrast, MS3-enriched signatures, including interferon responses and other inflammatory processes, correlate negatively with HSA21 cluster 1 genes and positively with HSA21 cluster 2 genes, as demonstrated by the relationship between *CXCR1* (encoded on chromosome 2) versus *PSMG1* (HSA21 cluster 1), and *KCNJ15* (HSA21 cluster 2) (Fig. 3f, g, i, Supplementary Fig. 3e).

Altogether, these findings indicate that the dysregulation of several key biological pathways in DS is variable and associated with differential expression patterns of HSA21 genes.

## Plasma proteomics reveals varied immune and inflammatory dysregulation across subtypes

To identify differences between subtypes in their plasma proteomes we utilized linear modeling of SomaScan plasma proteomics data obtained from 304 individuals with DS with matching transcriptome data and 103 euploid controls. These analyses revealed vast proteomic differences between each of the subtypes and euploid controls (Supplementary Fig. 4a, Supplementary Data 3), with hundreds of changes being unique to specific subtypes (Fig. 4a). Using results from linear regressions, we then utilized IPA of canonical pathways to assess differences in signaling pathways (Fig. 4b, Supplementary Data 3). In addition, given the potential for changes in the plasma proteome to reflect diseases and disruption of tissue function, we assessed IPA signatures of diseases and biological functions (Fig. 4c, Supplementary Data 3). When considering the full T21 cohort, IPA of canonical pathways suggested proteomic changes indicative of activation of the coagulation system and acute phase response signaling, as well as inhibition of intrinsic prothrombin activation, DHCR24 signaling, and LXR/RXR activation (Fig. 4b). Moreover, analysis of diseases and biological functions indicated activation of several immune/inflammatory related processes including immune mediated inflammatory disease, chronic inflammatory disorder, and infiltration by neutrophils (Fig. 4c). Interestingly, MS3 appears to be driving most of these proteomic changes, particularly activation of acute phase response signaling and enrichment of immune/inflammatory signatures (Fig. 4b, c). This is prominently demonstrated at the individual protein level by multiple factors related to inflammation and immune dysregulation (Fig. 4d). For example, the acute phase response proteins SAA1, SAA2, CRP, IL-6, and IL-1RN, all show a heightened degree of elevation in MS3 compared to the other subtypes (Fig. 4d, e, Supplementary Fig. 4b). This is also the case for many other unique factors, including, among others, LCN2 (lipocalin 2), a neutrophil gelatinase-associated protein linked to inflammation and acute kidney injury; PROK2 (prokineticin 2), a secreted protein involved in inflammation and immune response modulation; BPI

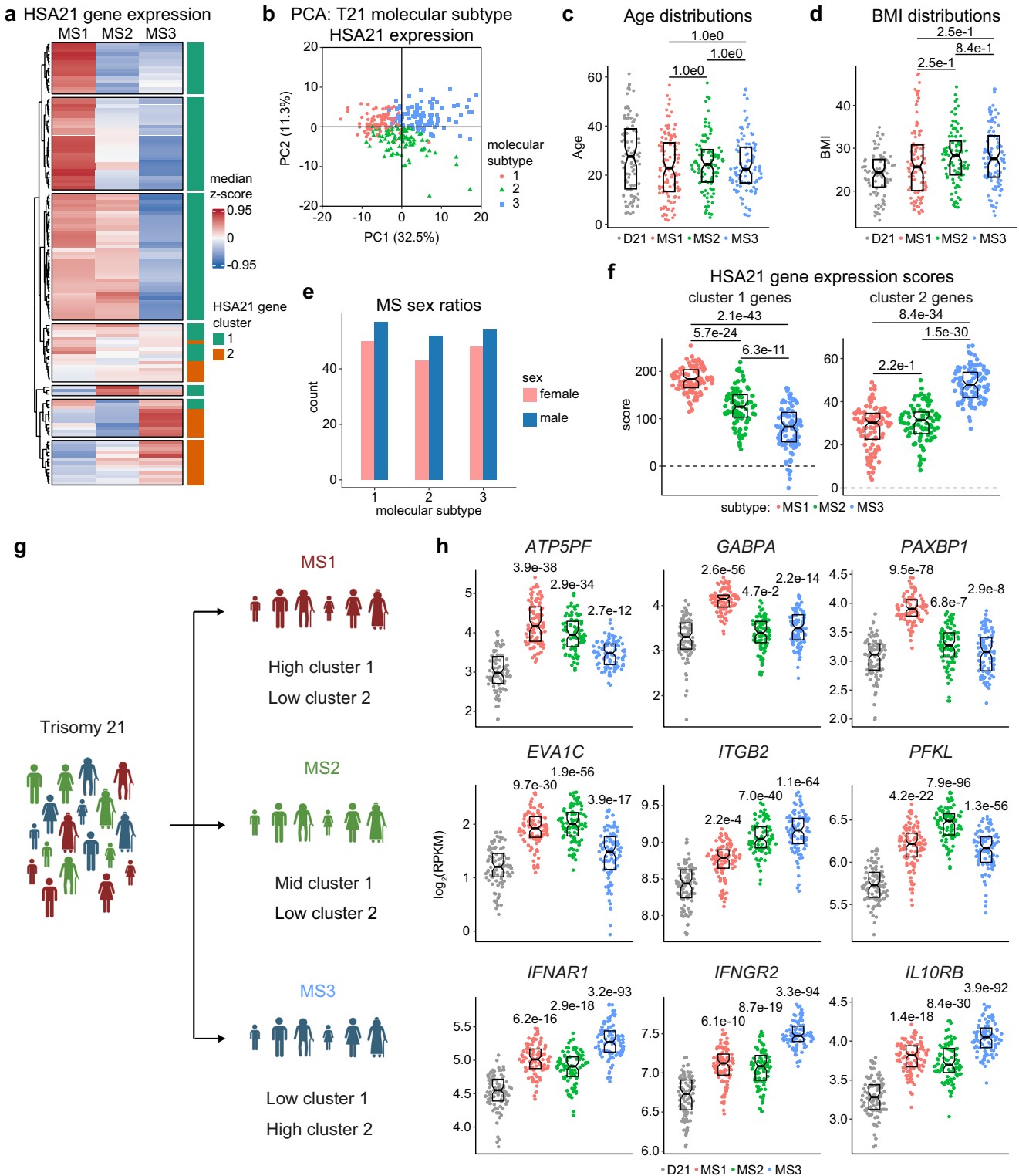

(bactericidal permeability increasing protein), a lipopolysaccharide binding protein found in neutrophils that plays a crucial role in the innate immune response; MPO (myeloperoxidase), an enzyme found in neutrophils and monocytes that attacks pathogens via production of reactive oxygen species; and PRTN3 (proteinase 3) a neutrophil-related enzyme involved in inflammation and a target antigen in autoimmune diseases like Wegener's granulomatosis (Fig. 4e, Supplementary Fig. 4b). As expected, when defining correlations of HSA21 gene expression vs. all proteins (Supplementary Fig. 4c), we found that factors uniquely elevated in MS3 positively correlate with

expression of HSA21 cluster 2 genes, but negatively correlate with HSA21 cluster 1 genes (Fig. 4f, Supplementary Fig. 4d). This is demonstrated most prominently by the relationship between levels of SAA1, PROK2, and BPI against expression of the HSA21 cluster 2 genes *KCNJ15* and *IFNGR2* (Fig. 4g).

Collectively, these results illustrate the heterogeneity of plasma proteomic changes in DS. Moreover, preferential overexpression of HSA21 cluster 2 genes, as seen in MS3, is associated with proteomic changes indicative of aberrant inflammatory- and immune-related processes.

**Fig. 2 | Variable chromosome 21 gene expression distinguishes molecular subtypes in Down syndrome. a** Heatmap showing unsupervised hierarchical clustering of HSA21 gene expression in trisomy 21 (T21) molecular subtype 1 (MS1) ($n = 107$), MS2 ($n = 95$) and MS3 ($n = 102$). Values are median z-scores across all T21 ($n = 304$). Colors indicate HSA21 gene cluster 1 (teal) and cluster 2 (orange). **b** Principal component analysis showing separation of subtypes based on HSA21 gene expression. **c, d** Sina plots showing similar age (**c**) and BMI (**d**) distributions between euploid controls (D21, $n = 96$, gray), MS1 ($n = 107$, red), MS2 ($n = 95$, green), and MS3 ($n = 102$, blue), with no significant differences as determined by two-sided Wilcoxon rank-sum tests. Statistics represent adjusted *p*-values (q-values) after Benjamini–Hochberg adjustment for multiple hypotheses. Boxes represent interquartile ranges and medians, with notches approximating 95% confidence intervals. **e** Bar charts showing similar sex ratios across subtypes. Colors indicate females (pink) and males (blue). **f** Sina plots showing HSA21 gene cluster polygenic scores in MS1 ($n = 107$, red), MS2 ($n = 95$, green), and MS3 ($n = 102$, blue). Scores were derived for each HSA21 gene cluster by calculating the sum of their z-scores relative to D21 ($n = 96$). Dashed line indicates D21 mean values. Boxes represent interquartile ranges and medians, with notches approximating 95% confidence intervals. Statistics indicate q-values from linear regressions, adjusted using the Benjamini–Hochberg method. **g** Illustration showing distinct molecular subtypes among individuals with T21, differentiated by unique overexpression profiles for HSA21 cluster 1 and 2 genes. **h** Sina plots showing expression of example HSA21 genes in D21 ($n = 96$, gray), MS1 ($n = 107$, red), MS2 ($n = 95$, green), and MS3 ($n = 102$, blue). Statistics above datapoint swarms are q-values, derived from DESeq2 comparisons to D21 and adjusted using the Benjamini–Hochberg method. Boxes represent interquartile ranges and medians, with notches approximating 95% confidence intervals. Panel **g** created with BioRender.com released under a Creative Commons Attribution-NonCommercial-NoDerivs 4.0 International license.

## Molecular subtypes associate with distinct inflammatory milieus

Prompted by the clear differences observed in the transcriptomic and proteomic immune signatures among subtypes, we employed an orthogonal approach and analyzed a select panel of immune markers using quantitative Meso Scale Discovery (MSD) targeted proteomics across 249 individuals with DS and 131 euploid controls (D21) with matched transcriptome data. When analyzing differences in MSD analyte levels, comparing the subtypes to euploid controls (D21) and to each other, both commonalities and differences among subtypes are observed (Fig. 5a, Supplementary Fig. 5a, b, Supplementary Data 4). For example, all three subtypes show significant elevation of IL-10, TSLP and TNF-alpha (Supplementary Fig. 5b, c). In contrast, relative to the other subtypes, MS3 displays significant elevation of IL-6 and downstream IL-6-inducible acute phase proteins CRP, SAA, and IL-1RN (Fig. 5b, Supplementary Fig. 5b, d), thus confirming our observations from SomaScan plasma proteomics of heightened acute phase response in MS3. Furthermore, MS3 also displays differential elevation of IL-22, IL-17A, and IL-15, interleukins involved in promoting inflammation, epithelial defense, and modulating T and NK cell activity (Supplementary Fig. 5d). Conversely, MS1 is the only subtype that shows significant upregulation of IL-8, eotaxin, and FGF basic, molecules involved in chemotaxis, allergic responses, and cell growth and differentiation, respectively (Supplementary Fig. 5e). These results demonstrate the subtypes of DS are distinguished by differences in their inflammatory milieu.

We then defined associations between HSA21 gene expression and levels of the immune markers, which demonstrated that whereas HSA21 cluster 1 gene expression correlates with cytokines preferentially elevated in MS1 (e.g., IL-8), HSA21 cluster 2 gene expression correlates with cytokines preferentially elevated in MS3 (e.g., IL6) (Fig. 5c). This is exemplified by the relationship between the HSA21 cluster 2 gene *KCNJ15* vs. levels of IL-6, CRP, and SAA (Fig. 5d).

Given that several inflammatory markers in the MSD panel are regularly assessed in the clinical setting (e.g., CRP, IL-6), we were interested to derive a cytokine/chemokine-based inflammatory metric that could differentiate subtypes. Toward this end we derived a multivariate score based on the immune marker profile of MS3. Specifically, we calculated the sum of z-scores, relative to euploid controls, for MSD analytes significantly elevated (fold change >1, $q < 0.1$, 10% FDR) in MS3 relative to euploid controls and MS1 (i.e., IL-6, CRP, SAA, IL-22, MIP-3alpha, IL-15, IL-1RN). We then analyzed differences in these scores across subtypes by Wilcoxon rank-sum test. This demonstrated that, on average, all subtypes show elevated cytokine scores relative to euploid controls (Fig. 5e). However, MS3 shows clear elevation relative to MS1 and MS2, which are not significantly different from each other (Fig. 5e). Expectedly, MS3 cytokine scores negatively and positively correlate, respectively, with expression of HSA21 cluster 1 genes and HSA21 cluster 2 genes (Fig. 5f). This is exemplified by relationships between MS3 cytokine scores against the HSA21 cluster 1 genes *TTC3* and *PRDM15*, and the

HSA21 cluster 2 genes *KCNJ15* and *IFNGR2* (Fig. 5g). When performing GSEA on the correlations between cytokine scores and all mRNAs in the RNA-seq dataset, we find that cytokine scores are associated with MS3-related transcriptomic signatures including interferon responses and other inflammatory pathways (Fig. 5h). Lastly, analyzing by IPA correlations against SomaScan proteins reveals MS3 cytokine scores track with factors involved in the acute phase response canonical pathway signature (Fig. 5i).

Collectively, these findings suggest the T21 subtypes are characterized by distinct profiles of cytokine and inflammatory markers, which align with their differentiating multiomic characteristics.

## Plasma metabolomics reveals depletion of amino acids associates with MS3

We next applied our analytical pipeline to plasma metabolomics data from 304 individuals with T21 with matching transcriptome data and 103 euploid controls (D21) using linear regression analyses, comparing each subtype to euploid controls and to each other (Supplementary Fig. 6a, Supplementary Data 5). This exercise revealed substantial differences in the plasma metabolomes of all subtypes compared to euploid controls. However, most changes were common among all subtypes, with few distinctions (Supplementary Fig. 6b). Comparison of MS3 to MS1 was the only analysis between subtypes showing differential abundance of metabolites, with 9 analytes being depleted in MS3 relative to MS1 (Supplementary Fig. 6a). These differential analytes encompassed the non-essential amino acids asparagine, serine, proline, and alanine (Fig. 6a, Supplementary Fig. 6c), essential amino acids threonine and histidine (Fig. 6b), and the amino acid metabolites L-citrulline and gamma-L-Glutamyl-D-alanine (Supplementary Fig. 6d). Notably, nearly all amino acids are depleted across the T21 subtypes relative to euploid controls, although most prominently for MS3 (Fig. 6c).

To explore potential drivers of amino acid depletion, we investigated expression of enzymes involved in biosynthesis of non-essential amino acids depleted in MS3. Indeed, expression of several key enzymes are downregulated in MS3 compared to MS1 including *ASNS* (Asparagine Synthetase), which catalyzes the conversion of aspartate to asparagine; *PSAT1* (Phosphoserine Aminotransferase 1), which dephosphorylates 3-phospho-L-serine to form serine; and *PYCR1* and *3* (Pyrroline-5-Carboxylate Reductase 1-3), which catalyze the conversion of pyrroline-5-carboxylate to proline (Fig. 6d, Supplementary Fig. 6e). This suggests depletion of non-essential amino acids in MS3 could be due in part to decreased expression of key enzymes involved in their synthesis.

Conversely, depletion of plasma amino acids could also be related to increased expression of transporters involved in cellular and tissue uptake of amino acids. Toward this end, we investigated expression of a panel of transporters ($n = 31$) involved in cellular and tissue uptake of amino acids. Of these transporters, 18 are significantly upregulated and 12 are significantly downregulated in MS3 relative to MS1. Notably, upregulated transporters include those

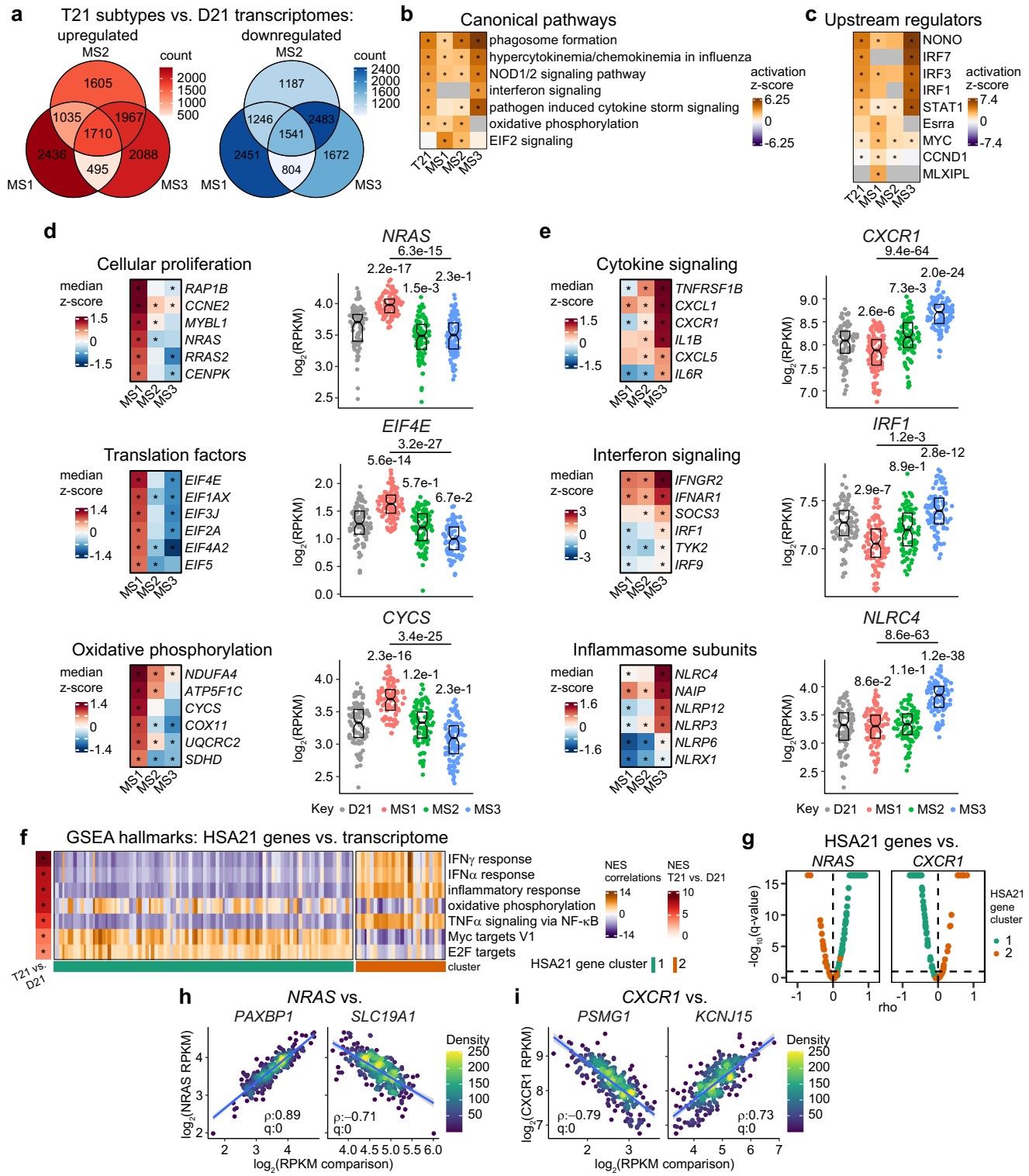

involved in uptake of histidine (*SLC15A3, SLC15A4, SLC66A1*), threonine (*SLC1A5*), alanine (*SLC38A10, SLC1A4*), serine (*SLC1A4*), and asparagine (*SLC1A5*) (Fig. 6e, Supplementary Fig. 6f). When defining correlations of HSA21 gene expression vs. metabolites (Supplementary Fig. 6g), we find that levels of amino acids depleted in MS3 negatively correlate with expression of HSA21 cluster 2 genes (Fig. 6f), as demonstrated by levels of asparagine vs. expression of *KCNJ15* and *MX2* (Supplementary Fig. 6h). Furthermore, except for proline, all amino acids depleted in MS3 are negatively correlated with the MS3 cytokine score, as demonstrated by correlations against asparagine levels (Fig. 6g, h).

Altogether, these results clearly demonstrate the inflammatory profile of MS3 is concurrent with depletion of plasma amino acids, which is potentially linked with decreased expression of biosynthetic enzymes and increased expression of uptake transporters.

## The molecular subtypes of Down syndrome present unique immune cell landscapes

Based on the observations that T21 molecular subtypes have distinct multiomics landscapes, we investigated differences with respect to their immune cell profiles. Toward this end, we performed beta regression analysis of mass-cytometry data from 278 individuals

**Fig. 3 | Molecular subtypes of Down syndrome are distinguished by distinct transcriptomic landscapes. a** Overlapping differentially expressed genes identified by DESeq2 in comparisons of each molecular subtype (MS1, $n = 107$; MS2, $n = 95$; MS3, $n = 102$) against euploid controls (D21, $n = 96$). **b, c** Heatmaps showing select canonical pathways (**b**) and upstream regulators (**c**) from Ingenuity Pathway Analysis (IPA) of DESeq2 results comparing trisomy 21 (T21, $n = 304$), MS1 ($n = 107$), MS2 ($n = 95$) and MS3 ($n = 102$) to D21 ($n = 96$). Asterisks denote $q < 0.1$ from IPA overrepresentation analysis. **d, e** Heatmaps display median z-scores for representative genes in MS1 ($n = 107$), MS2 ($n = 95$), and MS3 ($n = 102$) calculated relative to D21 ($n = 96$), with asterisks denoting $q < 0.1$ from DESeq2 after Benjamini–Hochberg adjustment. Sina plots illustrate gene expression across groups. q-values, derived from DESeq2 and adjusted using the Benjamini–Hochberg method, are displayed above individual data swarms for comparisons to D21 and above lines for MS3 vs. MS1. Boxes represent interquartile ranges and medians, with notches approximating 95% confidence intervals.

**f** Heatmap depicts pathway enrichment (NES, normalized enrichment score) from Gene Set Enrichment Analysis (GSEA) for HSA21 genes, based on their correlations against the transcriptome in individuals with T21 ($n = 304$). Left tile annotation shows results from GSEA of DESeq2 comparing T21 ($n = 304$) vs. D21 ($n = 96$). Asterisks denote $q < 0.1$ after Benjamini–Hochberg adjustment. Colors indicate HSA21 gene cluster 1 (teal) and cluster 2 (orange). **g** Volcano plots showing Spearman correlations of *NRAS* (left) and *CXCR1* (right) vs. HSA21 cluster 1 (teal) and cluster 2 (orange) genes in individuals with T21 ($n = 304$). Dashed line indicates $q = 0.1$. **h** Scatter plots showing expression of *NRAS* vs. *PAXBP1* (HSA21 gene cluster 1) and *SLC19A1* (HSA21 gene cluster 2) in individuals with T21 ($n = 304$). **i** Scatter plots showing expression of *CXCR1* versus *PSMG1* (HSA21 gene cluster 1) and *KCNJ15* (HSA21 gene cluster 2) in individuals with T21 ($n = 304$). For (**h, i**) *rho* and q-values (Benjamini–Hochberg adjusted *p*-values) for Spearman correlation are denoted. Points are colored by density; blue lines represent the fitted values from linear regressions, with 95% confidence intervals in grey.

with T21 having matching transcriptome data and 90 euploid controls (D21), with the goal of comparing the individual subtypes to euploid controls, which revealed variable dysregulation of multiple immune cell types (Fig. 7a, Supplementary Data 6). Among granulocytes, basophils are elevated in all three subtypes, concurrent with depletion of eosinophils, however, neutrophils are solely elevated in MS3 (Fig. 7a, b, Supplementary Fig. 7a). MS3 also displays the most prominent elevation of pro-inflammatory monocytes (i.e., intermediate, non-classical), along with increased frequencies of polymorphonucler myeloid-derived suppressor cells (PMN-MDSCs, Fig. 7a, c, Supplementary Fig. 7b). Interestingly, among lymphoid lineages, MS3 is the only subtype with a significantly decreased proportion of total T cells (Fig. 7a, d). Although all three subtypes show significant remodeling of the T cell lineage toward depletion of naive subsets and enrichment of effector and memory subsets, they also display some important differences, such as elevated frequencies of non-CD4+/CD8+T cells and CD8+TCM (T central memory) in MS3 (Fig. 7a, Supplementary Fig. 7c). All three subtypes display total B cell lymphopenia, although this effect is more pronounced in MS3 (Fig. 7a, e). Among B cell lineages, all subtypes show remodeling towards more terminally differentiated subsets such as mature B cells, IgM memory, and plasmablasts, along with increased frequencies of age-associated B cells (Fig. 7a). Among less differentiated or immature subsets, anergic B cells is the only subset showing a variable pattern, with depletion in both MS2 and MS3 (Supplementary Fig. 7d). Lastly, MS3 is the only group with significant increases in CD56+/CD16- NK cell subsets (Fig. 7a, Supplementary Fig. 7e).

When investigating the relationship between HSA21 gene expression and major immune cell subsets, we observed that cell populations enriched in MS3 (e.g., neutrophils) tend to be positively associated with expression of HSA21 cluster 2 genes, whereas cell types more depleted in MS3 (e.g., T cells, B cells) are associated with expression of HSA21 cluster 1 genes (Supplementary Fig. 7f). For example, whereas neutrophil frequencies are negatively associated with expression of the HSA21 cluster 1 gene *NDUFV3*, they are positively associated with expression of the HSA21 cluster 2 gene *PTTG1IP* (Fig. 7f). In contrast, T cells are positively associated with *NDUFV3* and negatively associated with *PTTG1IP* (Supplementary Fig. 7g).

In order to further assess hematological differences between the subtypes we analyzed complete blood count (CBC) parameters. Toward this end we calculated by linear regression differences between individual CBC measurements for each MS relative to euploid controls and to each other (Supplementary Data 6). This exercise revealed subtle differences between subtypes. Interestingly, all subtypes display stark signs of macrocytosis when compared to controls, as demonstrated by elevated mean corpuscular

volume (MCV) and mean corpuscular hemoglobin (MCH) (Fig. 7g). Furthermore, MS1 and MS3 show depletion of red blood cell counts (RBC), whereas MS1 and MS2 show total white blood cell (WBC) depletion (Fig. 7g). MS3 displays prominent depletion of lymphocytes, in both percent and absolute counts, in agreement with the observed T and B cell depletion observed by mass-cytometry (Fig. 7g, h).

Altogether, these results reveal distinct variations in the cellular composition of the peripheral blood cell populations among subtypes, which could contribute to their heterogenous multiomics landscapes.

## Temporal stability and clinical detection of molecular subtypes of Down syndrome

Next, we decided to evaluate to what degree the heterogenous presentation of molecular and immune subtypes of DS was conserved over time. Thus, we analyzed correlations of MSD immune markers in samples taken at least a year apart from a cohort ($n = 25$) of research participants with T21. This effort revealed that whereas some immune markers are highly stable from one year to another (e.g., CRP, IL12/23p40, IP-10, IL-31), others are not (e.g., bFGF, IL-17B, IL-4) (Fig. 8a, Supplementary Fig. 8a, Supplementary Data 7). Within this context, immune markers elevated in MS3 (i.e., CRP, SAA, IL-1RN, MIP-1α, IL-6, IL-15, and IL-22) remained relatively consistent leading to significantly stable MS3 cytokine scores over time (Fig. 8a–c). We performed similar repeat measurements on CBC parameters in a subset ($n = 33$) of individuals with T21 (Fig. 8d, Supplementary Data 7). This revealed multiple analytes to be stable over time including platelet counts, MCV, lymphocyte parameters (absolute and percent), and total white blood cells (Fig. 8d, e, Supplementary Fig. 8b). Collectively, these analyses show that clinically relevant molecular characteristics of MS3, such as heightened acute phase response factors and lymphopenia, are relatively stable over time. Prompted by these results, we aimed to define clinical indexes to classify the subtypes using temporally stable analytes distinguishable in MS3. Toward this end we calculated ratios of inflammatory markers in the IL-6 signaling pathway (i.e., IL-6, CRP, SAA) and neutrophils and lymphocyte measurements. In order to ensure the greatest sample overlap and statistical power, we utilized SomaScan measurements for inflammatory markers and mass-cytometry data for neutrophils. Expectedly, these ratios are consistently higher in MS3 (Fig. 8f). Using the 90th percentile values of these ratios in euploid controls, we were able to arrive to classifiers for MS3 with >90% sensitivity and specificity ranging from 48.8–67.5% (Fig. 8g). However, when combining multiple indexes into the classifier, specificity increased above 80%. Although the research measurements used for these classifiers are not certified for clinical use, these results nevertheless support the notion that molecular and

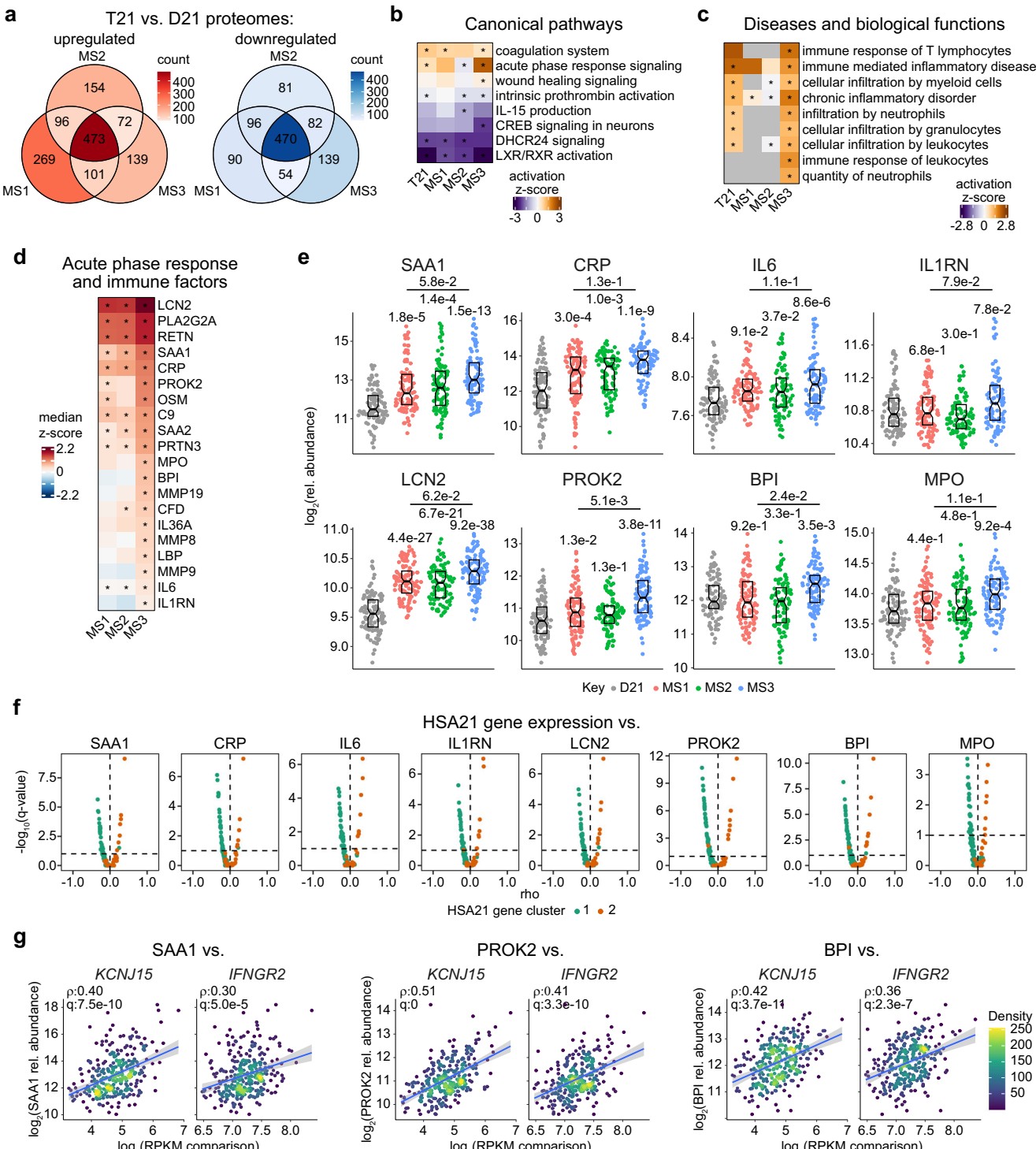

**Fig. 4 | Plasma proteomics reveals varied immune and inflammatory dysregulation across subtypes. a** Diagrams show overlapping differentially abundant proteins identified by linear regressions in comparisons of each molecular subtype (MS1, *n* = 107; MS2, *n* = 95; MS3, *n* = 102) against euploid controls (D21, *n* = 103). **b**, **c** Heatmaps showing select canonical pathways (**b**) and disease and biological functions (**c**) from Ingenuity Pathway Analysis (IPA) of linear regression results comparing trisomy 21 (T21, *n* = 304), MS1 (*n* = 107), MS2 (*n* = 95) and MS3 (*n* = 102) to D21 (*n* = 103), with asterisks denoting q < 0.1 from IPA overrepresentation analysis **d** Heatmap showing median z-scores relative to euploid controls (D21, *n* = 103) for representative inflammatory and immune-related proteins in MS1 (*n* = 107), MS2 (*n* = 95), and MS3 (*n* = 102). Asterisks indicate significance (q < 0.1) from linear regressions vs. D21 after Benjamini–Hochberg adjustment for multiple hypotheses. **e** Sina plots showing levels of example proteins from (**d**) in D21 (*n* = 103, gray),

MS1 (*n* = 107, red), MS2 (*n* = 95, green), and MS3 (*n* = 102, blue). q-values, derived from linear regressions and adjusted using the Benjamini–Hochberg method, are displayed above individual data swarms for comparisons to D21 and above lines for MS3 vs. MS1. Boxes represent interquartile ranges and medians, with notches approximating 95% confidence intervals. **f** Volcano plots showing Spearman correlations of proteins from (**e**) vs. mRNA expression of HSA21 cluster 1 genes (teal) and HSA21 cluster 2 genes (orange) in individuals with T21 (*n* = 304). Dashed line indicates q = 0.1. **g** Scatter plots depicting relationships between levels of SAA1, PROK2 and BPI proteins vs. expression of HSA21 cluster 2 genes *KCNJ15* and *IFNGR2* in individuals with T21 (*n* = 304). *rho* and q-values (Benjamini–Hochberg adjusted *p*-values) for Spearman correlation are denoted. Points are colored by density; blue lines represent the fitted values from linear regressions, with 95% confidence intervals in grey.

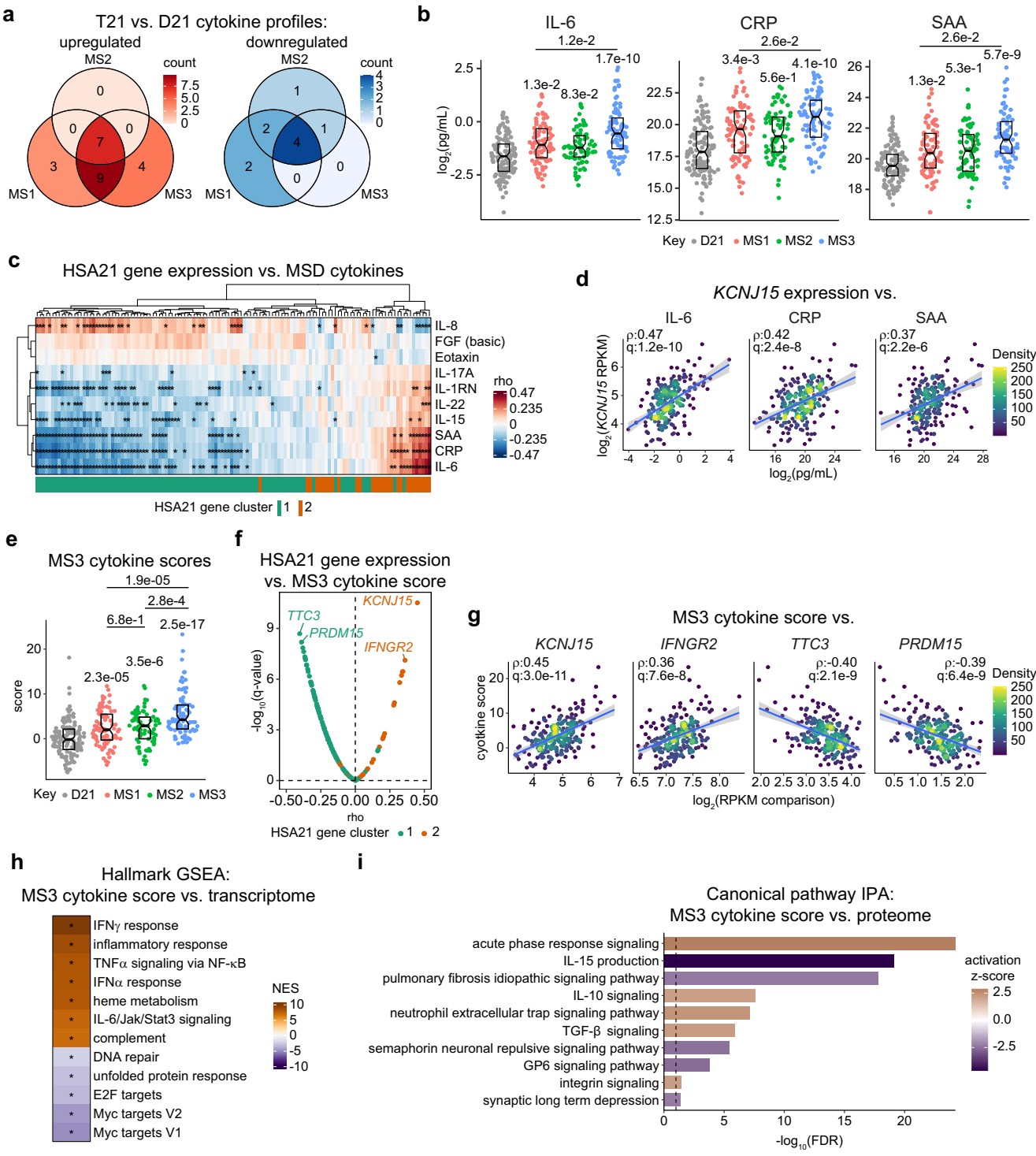

immune heterogeneity of DS could be gauged with commonly used clinical parameters.

Lastly, we investigated whether the molecular subtypes associated with different patterns of co-occurring conditions. This analysis revealed only modest trends in the overrepresentation of certain co-occurring conditions within specific subtypes, which did not achieve statistical significance after multiple hypotheses correction (Supplementary Fig. 8c). Nonetheless, these trends justify further investigations using larger cohorts to understand how variation in HSA21 gene expression may modulate the appearance and severity of various developmental and clinical hallmarks of DS.

## Discussion

DS manifests as a highly heterogeneous medical condition in terms of number and severity of phenotypes and co-occurring conditions[1]. This inherent heterogeneity poses significant challenges to the clinical management of DS, revealing the need for a personalized medicine approach to serve this population. A plausible driver of phenotypic variation in DS is inter-individual variability in HSA21 gene overexpression, as individuals with heightened or dampened overexpression of specific HSA21 genes may display unique characteristics from those with dissimilar expression profiles. However, deciphering the precise contributions of individual HSA21 genes to

**Fig. 5 | Molecular subtypes associate with distinct inflammatory milieus.**
**a** Overlapping differentially abundant inflammatory markers identified by linear regressions comparing each molecular subtype (MS1, *n* = 87; MS2, *n* = 75; MS3, *n* = 87) against euploid controls (D21, *n* = 131). **b** Sina plots for IL-6, CRP, and SAA in D21 (*n* = 131, gray), MS1 (*n* = 87, red), MS2 (*n* = 75, green), and MS3 (*n* = 87, blue). q-values (Benjamini–Hochberg adjusted *p*-values), derived from linear regressions are displayed above swarms for comparisons to D21 and above lines for MS3 vs. MS1. Boxes represent interquartile ranges (IQR) and medians, notches approximate 95% confidence intervals. **c** Clustering of Spearman correlations between HSA21 gene expression vs. inflammatory markers. Asterisks indicate q < 0.1 after Benjamini–Hochberg adjustment. Colors indicate HSA21 gene cluster 1 (teal) and 2 (orange). **d** Scatter plots depicting expression of *KCNJ15* vs. IL-6, CRP, and SAA in T21 (*n* = 249). *rho* and q-values (Benjamini–Hochberg adjusted *p*-values) for Spearman correlations are denoted. Points are colored by density; blue lines represent the fitted values from linear regressions, with 95% confidence intervals in

grey. **e** Cytokine scores in D21 (*n* = 131, gray), MS1 (*n* = 87, red), MS2 (*n* = 75, green) and MS3 (*n* = 87, blue). q-values (Benjamini–Hochberg adjusted *p*-values), derived from linear regressions are displayed above swarms for comparisons to D21. q-values over lines indicate comparisons between MS. Boxes represent IQRs and medians, notches approximate 95% confidence intervals. **f** Volcano plot showing Spearman correlations between cytokine scores vs. HSA21 gene expression in T21 (*n* = 249). Colors indicate HSA21 gene cluster 1 (teal) and 2 (orange). Dashed line indicates q = 0.1. **g** Scatter plots depicting examples from (**f**). *rho* and q-values (Benjamini–Hochberg adjusted *p*-values) for Spearman correlation are denoted. Points are colored by density; blue lines represent the fitted values from linear regressions, with 95% confidence intervals in grey. **h** Heatmap showing results from gene set enrichment analysis (GSEA) of correlations between cytokine scores and all mRNAs, with asterisks indicating q < 0.1 after Benjamini–Hochberg adjustment. **i** Results from canonical pathway Ingenuity Pathway Analysis (IPA) of correlations between cytokine scores and plasma proteins.

the varied phenotypes of DS is an ongoing challenge. Although some genes on HSA21 have been shown to contribute to specific traits of DS, such as *APP*[8], the interferon receptors[9], and *DYRK1A*[10], a comprehensive mapping of HSA21 gene-to-phenotype relationships is missing.

Whole blood transcriptome analysis demonstrates that HSA21 genes are not coordinately overexpressed in DS. We identified two distinct clusters of co-expressed genes on HSA21 and revealed three subtypes of individuals with T21, distinguished by their expression patterns of HSA21 genes (Fig. 9a, b). Several studies previously reported variability in HSA21 gene expression patterns in DS[13–16], and some studies proposed the existence of gene dosage compensation in DS, with both supporting and opposing evidence in the literature[14,16–21]. Collectively, we did not observe clear signs of global dosage compensation, with overexpression of HSA21 genes averaging the expected 1.5-fold increase. However, we observed strong inter-individual variability in HSA21 gene overexpression, revealing the presence of mechanisms differentially affecting distinct sets of genes on HSA21, such as epigenetic regulation or cell type-specific expression within the blood cell lineages. Epigenetic control is influenced by genetic variation, physiological changes, environmental exposures, and other mechanisms[22,23] and the expression of several HSA21 genes have been reported to be epigenetically regulated[24–26]. Cell type-specific variation in gene expression across blood cell types is also well documented, to the point that immune cell frequencies can be deduced, to some extent, from deconvolution of bulk transcriptome data[27,28]. Subsequent research should probe further into these potential mechanisms of variegated HSA21 overexpression, emphasizing potential regulons that could underlie the variability observed across subtypes.

Our analysis revealed that the whole blood transcriptome of MS1 is characterized by signatures related to cellular proliferation (e.g., MYC), mitochondrial metabolism (e.g., oxidative phosphorylation) and protein translation (e.g., EIF2 signaling), whereas MS3 presents a prominent pro-inflammatory phenotype, with elevated signatures of interferon signaling and other immune processes (Fig. 9c). Although all these signatures are elevated in the entire T21 cohort relative to euploid controls, it is now clear that distinct subsets of individuals within the DS population are driving these broader transcriptomic shifts, revealing an unexpected molecular heterogeneity. Considering the differential expression profiles of HSA21 genes across subtypes, it is plausible that inter-individual variability in HSA21 gene expression underlies this overall heterogeneity in transcriptome signatures. For example, heightened expression of the interferon receptors is likely contributing to the inflammatory and immune phenotype in MS3. In fact, we previously showed that normalization of interferon receptor copy number in the Dp16 mouse model of DS rescues hyperactive inflammatory transcriptomic signatures and several DS-associated phenotypes

including immune hypersensitivity, congenital heart defects, craniofacial abnormalities, developmental delays, and cognitive deficits[9]. Whether heightened expression of specific HSA21 cluster 1 genes is a driver of broader transcriptomic shifts characteristic of MS1 remains to be elucidated. Hence, future studies should be focused on understanding the roles of different HSA21 genes in influencing transcriptomic variability and overall clinical heterogeneity in T21.

While all subtypes exhibited increased levels of plasma inflammatory markers compared to the euploid population, each was characterized by a distinct proteomic profile. This not only further emphasizes the molecular heterogeneity of DS, but also suggests a potential influence of selective overexpression of specific HSA21 genes in shaping this variability. Most notably, the MS3 plasma proteome is marked by an amplified acute phase response, highlighted by elevated markers of IL-6 signaling (e.g., IL-6, CRP, SAA) (Fig. 9d). In contrast, the acute phase response signature is dampened in MS2 relative to euploid controls, and it is not different in MS1. Although these findings deepen our understanding of the molecular variation in T21, how this diversity might drive clinical heterogeneity remains to be determined. Furthermore, pinpointing individual, or sets of, HSA21 genes that drive these alterations is an ongoing effort. Elevated levels of both IL-6 and CRP have been previously reported in DS[29–32]. The clinical importance of these markers is underscored by their associations with a range of adverse clinical outcomes in the general population including autoimmunity[33,34], increased risk of cardiac complications (e.g., heart disease, failure, infarction)[35–37], cognitive impairment/decline[38–40], and compromised lung function[41–44]. As research progresses, future efforts should seek to deconvolute the relationship between variability in HSA21 gene overexpression, the resulting dysregulated proteomic signatures, and the broader impact on the clinical manifestations of DS.

Analysis of plasma metabolomics revealed that individuals with DS exhibit depletion of circulating amino acids, which was notably exacerbated in MS3 (Fig. 9e). Complementing these findings, results from whole blood RNA-seq showed decreased expression of enzymes essential for amino acid synthesis and increased expression of uptake transporters in MS3. However, it is unclear whether these metabolic differences arise from differential overexpression of HSA21 cluster 2 genes in MS3, its distinct inflammatory profile, or neither. Lending credence to the potential role of inflammation, amino acid catabolism is known to escalate amid inflammatory processes and both adaptive and innate immune responses[45]. Notably, we observed that amino acid levels negatively correlated with the MS3 cytokine score. In addition, various amino acid transporters upregulated in our analysis have been shown to play roles in immunological responses. For example, *SLC1A5* is induced by IL-2 and is necessary for interferon gamma production and degranulation[46,47]; *SLC7A5* is increased in patients with rheumatoid arthritis and is induced in monocytes by lipopolisacharide and

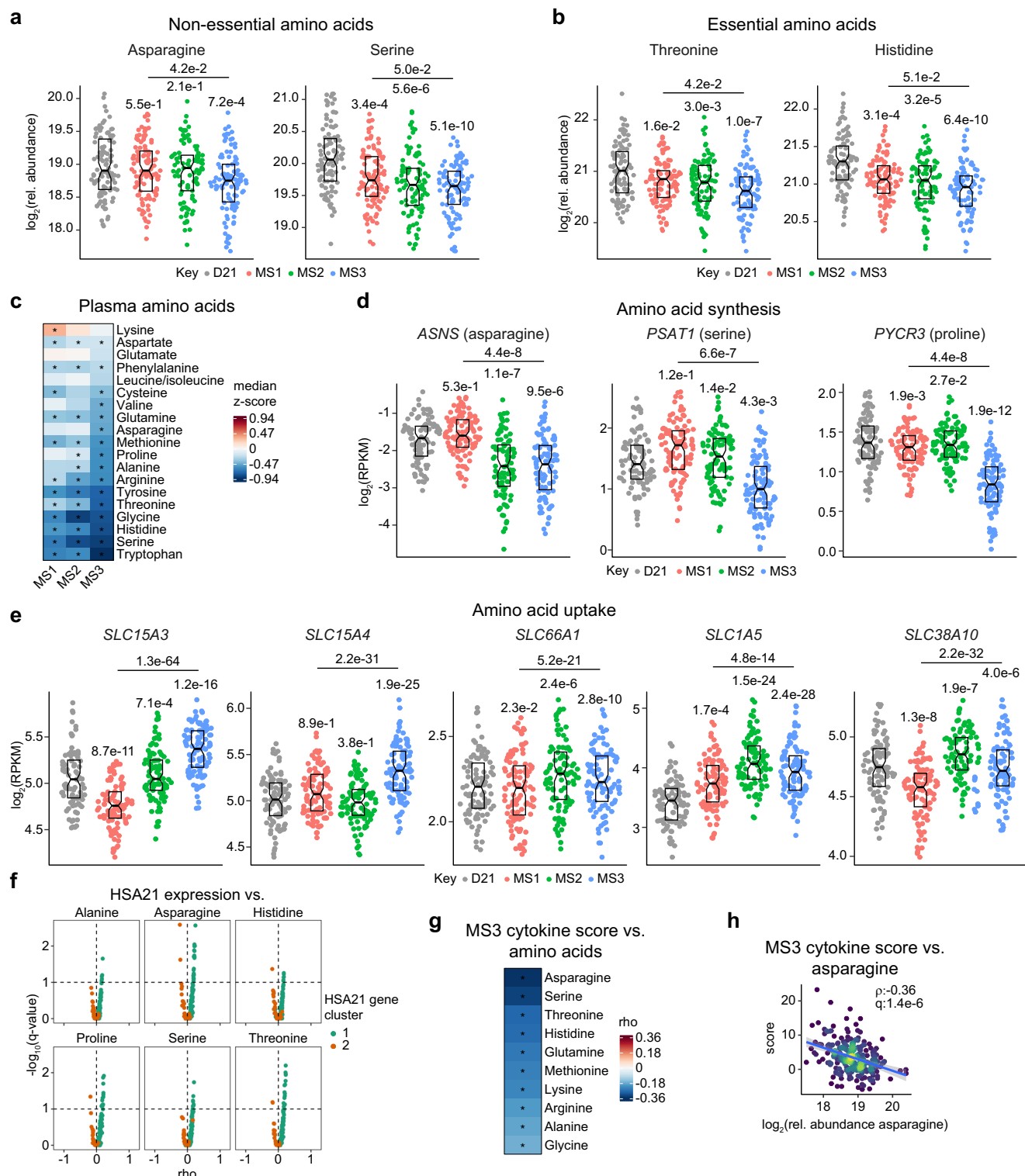

infection[48]; *SLC15A4* is necessary for TLR-dependent type I interferon production and is involved in IL-1B production[49]; and *SLC15A3* is upregulated by TLR2, 4, 7 and 9 via NfkB activation[50]. Notably, individuals with DS display traits of sarcopenia and hypotonia[51,52], which can affect circulating amino acid levels. Hence, future investigations should seek to unravel the relationships between HSA21 gene overexpression, amino acid depletion, and clinical conditions associated with DS.

We observed that all three subtypes exhibit some degree of immune cell alterations (Fig. 9f). However, MS3 displayed the

most pronounced and distinctive changes characteristic of DS. This was marked by significant enrichment of neutrophil and inflammatory monocyte populations, coupled with stark depletion of both T cells and B cells. While all subtypes displayed noticeable depletion of naive T and B subsets, a pattern consistent with previous literature in DS[53–55], MS3 alone shows total T cell lymphopenia and the most pronounced B cell lymphopenia. CBC analysis unequivocally supported the finding of pronounced lymphopenia in MS3. These significant immune shifts observed in MS3, alongside its selective overexpression of HSA21 cluster 2

**Fig. 6 | Plasma metabolomics reveals depletion of amino acids associates with MS3. a, b** Sina plots showing levels of non-essential amino acids (**a**) and essential amino acids (**b**) in MS1 ($n = 107$, red), MS2 ($n = 95$, green), and MS3 ($n = 102$, blue) relative to euploid controls (D21, $n = 103$, gray). q-values, derived from linear regressions and adjusted using the Benjamini–Hochberg method, are displayed above individual data swarms for comparisons to D21 and above lines for MS3 vs. MS1. Boxes represent interquartile ranges and medians, with notches approximating 95% confidence intervals. **c** Heatmap showing median z-scores relative to euploid controls (D21, $n = 103$) of all amino acids in metabolome dataset across MS1 ($n = 107$), MS2 ($n = 95$), and MS3 ($n = 102$). Asterisks indicate significance (q < 0.1) from linear regressions vs. D21 after Benjamini–Hochberg adjustment for multiple hypotheses. **d, e** Sina plots showing mRNA expression of enzymes related to non-essential amino acid biosynthesis (**d**) and tissue uptake (**e**) in D21 ($n = 103$, gray), MS1 ($n = 107$, red), MS2 ($n = 95$, green), MS3 ($n = 102$, blue). q-values, derived

from DESeq2 and adjusted using the Benjamini–Hochberg method, are displayed above individual data swarms for comparisons to D21 and above lines for MS3 vs. MS1. Boxes represent interquartile ranges and medians, with notches approximating 95% confidence intervals. **f** Volcano plots showing Spearman correlations between levels of amino acids vs. expression of HSA21 cluster 1 (teal) and HSA21 cluster 2 (orange) genes in individuals with T21 ($n = 304$). Dashed line indicates q = 0.1. **g** Heatmap showing Spearman correlations between cytokine scores and levels of amino acids. Asterisks indicate significance (q < 0.1) after Benjamini–Hochberg adjustment for multiple hypotheses. **h** Scatter plots depicting relationship between cytokine scores vs. asparagine levels in individuals with T21 ($n = 304$). *rho* and q-values (Benjamini–Hochberg adjusted *p*-values) for Spearman correlation are denoted. Points are colored by density; blue lines represent the fitted values from linear regressions, with 95% confidence intervals in grey.

genes and its unique inflammatory landscape raise questions about the interplay between genetic and immune factors in DS. Given these changes, understanding the relationship between these immune alterations and other DS-related clinical manifestations will be a key aspect of future research endeavors. The exact mechanisms underlying the pronounced lymphopenia observed in MS3, and whether it stems from developmental issues, active depletion, or a combination of both, needs further elucidation.

While this study provides insights into the molecular heterogeneity of T21, there are some limitations to be considered. First, it is unclear whether these MS classifications are stable or if they change over time. This was partly addressed by longitudinal analyses of cytokine and CBC measurements which showed characteristics of MS3 are stable over a one-year period. Future investigations could seek to investigate stability of the multiomics signatures that define each subtype over longer periods of time. Another limitation is the lack of a clear relationship between the molecular features of each subtype and their clinical variables. We observed that the three subtypes are not significantly different in terms of age, sex distribution, or BMI, but whether or not they show significant differences in terms of number and/or severity of core DS phenotypes and co-occurring conditions will require further investigation and larger sample sizes. These future investigations would not only enhance our understanding of how the observed molecular and immune subtypes influence pathological processes in DS but may also provide insights toward tailored therapeutic applications for the different T21 subtypes.

In this study, we have established that HSA21 gene overexpression is not uniformly exhibited across individuals with DS. This finding challenges a prevailing notion of consistent overexpression of all HSA21 genes in DS, revealing instead that the extent of overexpression varies both among different HSA21 genes and across individuals. This variability, as demonstrated through the identification of the molecular subtypes, seems to underpin distinct downstream molecular profiles, provoking intriguing questions about the nature of T21 pathobiology. It is unclear at this point whether MS1-3 represent stable and distinct pathophysiological states versus variations along a continuum. While some of our findings place MS2 as an intermediate along a spectrum between MS1 and MS3 (e.g., HSA21 gene cluster 1 expression, plasma amino acid depletion), it also exhibits unique traits not seen in the other subtypes (e.g., signatures of inhibited acute phase response). This intricate scenario underscores the need for deeper exploration, particularly through integrating clinical outcomes and data on co-occurring conditions. Future research that links these molecular differences with clinical manifestations in DS will enhance our understanding of the diverse pathobiology of DS, paving the way for personalized care strategies tailored to the unique molecular landscape of individuals with DS.

## Methods

### Study participants

The results and analyses presented herein are part of a nested study within the Crnic Institute's Human Trisome Project (HTP) cohort study. Participants in the HTP were enrolled under a protocol approved by the Colorado Multiple Institutional Review Board (COMIRB 15-2170, NCT02864108, see also www.trisome.org). All study participants, or their legal guardians, provided written informed consent. In addition to biospecimen collection, a clinical history for each participant was curated from both medical records and participant/family reports. Medical records took precedence in cases of discordance. Biological datasets analyzed in this study were generated from de-identified biospecimens and linked to demographic and clinical metadata. All research participants consented for sharing of their de-identified demographics and clinical metadata.

### Inclusion and ethics

This study included local researchers throughout the entire research process. This research is relevant both locally and globally. This has been determined in collaboration with local advocacy organizations supporting the population with Down syndrome. The roles and responsibilities of the authors and collaborators were determined both ahead of research and as needed throughout the process. This research was not restricted or prohibited in the setting where it was conducted. Participants were enrolled under a protocol approved by the Colorado Multiple Institutional Review Board (COMIRB 15-2170, NCT02864108, see also www.trisome.org). While this research does not result in stigmatization, incrimination, discrimination, or otherwise personal risk to participants, all information has been de-identified to ensure the safety and wellbeing of research participants. All research activities and experimental procedures were carried out with strict compliance with all federal, state, local and University regulations to mitigate any potential risks to the collaborators. Biospecimens utilized in this research are available through the Human Trisome Project Biobank and can be requested online through www.trisome.org.

### Blood sample collection and processing

The biological datasets analyzed herein were derived from peripheral blood samples and matched to demographic and clinical metadata. Peripheral blood was collected using PAXgene RNA Tubes (Qiagen) and BD Vacutainer K2 EDTA tubes (BD). Whole blood from PAXgene collection tubes was processed for RNA sequencing as described below. Samples collected from EDTA tubes were processed and used to generate other biological datasets as described. Two 0.5 mL aliquots of whole blood were withdrawn from each tube and processed for mass cytometry as described below. The remaining EDTA blood samples were centrifuged at $700 \times g$ for 15 min to separate plasma, buffy coat containing white blood cells

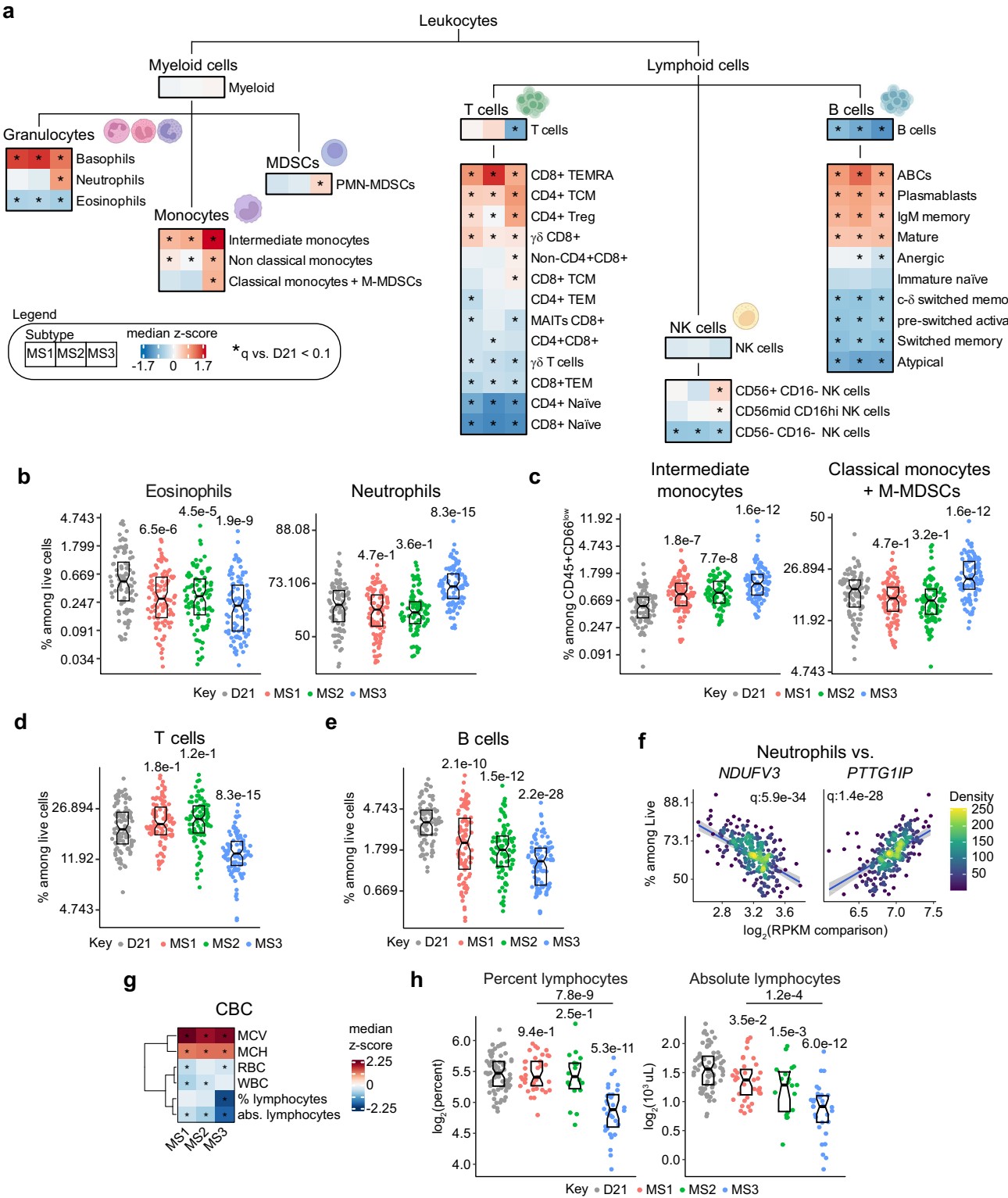

(WBC), and red blood cells (RBCs). Samples were then aliquoted, flash frozen and stored at −80 °C until subsequent processing and analysis. Centrifugation and storage of samples took place within 2 h of collection.

**Whole-blood RNA sequencing**

RNA was extracted and purified from whole blood collected in PAXgene RNA tubes using a PAXgene Blood RNA Kit (PreAnalytiX). RNA extracts were assessed for quality using a 2200 TapeStation system (Agilent) and quantified on a Qubit fluorometer (Invitrogen). Globin RNA depletion was carried out using a GlobinClear kit (ThermoFisher Scientific). Poly-A( + ) RNA enrichment and strand-specific library preparation was carried out using NEBNext Poly(A) mRNA Magnetic Isolation Module, and NEBNext Ultra II Directional RNA Library Prep Kit for Illumina (New England Biolabs). Paired-end sequencing with a read length of 150 bp was carried out by Novogene Co., Ltd on an Illumina NovaSeq 6000 instrument. Files were delivered in FASTQ format.

**Fig. 7 | The molecular subtypes of Down syndrome present unique immune cell landscapes. a** Illustration of immune cell differentiation and heatmaps comparing levels of individual immune cell types in MS1 (*n* = 98), MS2 (*n* = 90), MS3 (*n* = 96). Values are median z-scores relative to euploid controls (D21, *n* = 96). Asterisks indicate significance (q < 0.1) from beta regressions vs. D21 after Benjamini−Hochberg adjustment for multiple hypotheses. **b**–**e** Sina plots showing levels of eosinophils and neutrophils (**b**), intermediate monocytes and classical monocytes (**c**), T cells (**d**), and B cells (**e**) in MS1 (*n* = 98, red), MS2 (*n* = 90, green), MS3 (*n* = 96, blue) compared to euploid controls (D21, *n* = 90, gray). q-values, derived from beta regressions and adjusted using the Benjamini−Hochberg method, are displayed above individual data swarms for comparisons to D21. Boxes represent interquartile ranges and medians, with notches approximating 95% confidence intervals. **f** Scatter plots depicting relationship between neutrophil levels vs. expression of HSA21 cluster 1 gene (*NDUFV3*) and HSA21 cluster 2 gene (*PTTG1IP*). Points are colored by density; blue lines represent the fitted values from

beta regressions, with 95% confidence intervals in grey. Significance is defined as q < 0.1 after multiple hypothesis correction using the Benjamini−Hochberg method. **g** Heatmap showing levels of complete blood count (CBC) parameters in MS1 (*n* = 39), MS2 (*n* = 19), MS3 (*n* = 33). Values are median z-scores relative to euploid controls (D21, *n* = 70). Asterisks indicate significance (q < 0.1) from linear regressions vs. D21 after Benjamini−Hochberg adjustment for multiple hypotheses. **h** Sina plots showing levels for lymphocyte percentage and absolute lymphocytes in MS1 (*n* = 39, red), MS2 (*n* = 19, green), MS3 (*n* = 33, blue) compared to euploid controls (D21, *n* = 70, gray). q-values, derived from linear regressions and adjusted using the Benjamini−Hochberg method, are displayed above individual data swarms for comparisons to D21 and above lines for MS3 vs. MS1. Boxes represent interquartile ranges and medians, with notches approximating 95% confidence intervals. Panel **a** created with graphical elements from BioRender.com released under a Creative Commons Attribution-NonCommercial-NoDerivs 4.0 International license.

## SomaScan® plasma proteomics

EDTA plasma (125 µL) was analyzed by SomaScan® using established protocols[56]. Briefly, SOMAmer reagents (*n* = 4500+) bind target peptides in the sample and levels are quantified on custom Agilent hybridization chips. Normalization and calibration were performed according to SomaScan Data Standardization and File Specification Technical Note (SSM-020)[56]. The output of the SomaScan assay is reported in relative fluorescent units (RFU).

## Profiling of plasma immune markers using Meso Scale Discovery assays

Analysis of plasma immune markers was performed as previously described[11]. Briefly, from each EDTA plasma sample, two replicates of 12–25 µL were analyzed using the Meso Scale Discovery (MSD) multiplex immunoassay platform V-PLEX Human Biomarker 54-Plex Kit (Cat # K15248D) on a MESO QuickPlex SQ 120 instrument. Assays were carried out as per the manufacturer's instructions. Concentration values were calculated against a standard curve with provided calibrators. MSD data are reported as concentration values in picograms per milliliter of plasma.

## Mass cytometry of white blood cells

For mass cytometry, two 0.5 mL aliquots of EDTA whole blood samples underwent red blood cell lysis and white blood cell fixation using Lyse/Fix Buffer (BD Phosflow Lyse/Fix Buffer 5X, BD Biosciences). White blood cells were then washed 1x in PBS (Rockland), resuspended in Cell Staining Buffer (Fluidigm) and stored at −80 °C. For antibody staining, samples were thawed at room temperature, washed in Cell Staining Buffer, barcoded using a Cell-ID 20-Plex Pd Barcoding Kit (Fluidigm), and combined per batch. Each batch was able to accommodate 19 samples with a common reference sample. Antibodies were either purchased pre-conjugated to metal isotopes or conjugation was performed in-house using a Maxpar Antibody Labeling Kit (Fluidigm). See Supplementary Data 8 for antibody details. Working dilutions for antibody staining were titrated and validated using the common reference sample and comparison to relative frequencies obtained by independent flow cytometry analysis. Surface marker staining was carried out for 30 min at 4 °C in Cell Staining Buffer with added Fc Receptor Binding Inhibitor (eBioscience/ThermoFisher Scientific). Staining was followed by a wash in Cell Staining Buffer. Next, cells were permeabilized in Buffer III (Transcription Factor Phospho Buffer Set, BD Pharmingen) for 20 min at 4 °C followed by washing with perm/wash buffer (Transcription Factor Phospho Buffer Set, BD Pharmingen). Intracellular transcription factor and phospho-epitope staining was carried out for 1 h at 4 °C in perm/wash buffer (Transcription Factor Phospho Buffer Set, BD Pharmingen), followed by a wash with Cell Staining Buffer. Cell-ID Intercalator-Ir (Fluidigm) was used to label barcoded and stained cells. Labeled cells were analyzed on a Helios instrument

(Fluidigm). Mass cytometry data were exported as v3.0 FCS files for pre-processing and analysis.

## Mass spectrometry-based plasma metabolomics and lipidomics

EDTA plasma samples were thawed on ice and extracted using a modified Folch method (chloroform/methanol/water 8:4:3) as previously described[11]. Briefly, samples (20 µL) were diluted in water (130 µL, LC-MS grade), a 2:1 mixture of chloroform/methanol was added (600 µL), and the mixture was vortexed for 10 s. This was followed by incubation at 4 °C for 5 minutes, a pulse vortex (5 s), then centrifugation (14,000 × *g* for 10 min) at 4 °C. The aqueous (top) and organic (bottom) phases were transferred to separate tubes for metabolomic (aqueous) and lipidomic (organic) analysis. Analyses were carried out using a Vanquish UHPLC coupled online to a Q Exactive high resolution mass spectrometer (ThermoFisher Scientific). As previously described[57], using a 5 min C18 gradient on a Kinetex C18 column (Phenomenex), samples (10 µL per injection) were randomized and analyzed using positive and negative electrospray ionization methods in separate runs. Data were analyzed using Maven[58], the KEGG database and an in-house standard library.

## Complete blood count assessments

Complete blood count (CBC) measurements were performed using whole blood samples collected in EDTA tubes. Prior to analysis, samples were thoroughly mixed to ensure uniformity. A volume of 12 µL was then processed using an AcT 10 hematology analyzer (Beckman Coulter). Results are reported as white blood cell count (WBC, $10^3$/µL), red blood cell count (RBC, $10^3$/µL), hemoglobin (Hgb, g/dL), hematocrit (Hct, %), mean corpuscular volume (MCV, fL), mean corpuscular hemoglobin (MCH, pg), mean corpuscular hemoglobin concentration (MCHC, g/dL), platelet count (Plt, $10^3$/µL), absolute lymphocyte count ($10^3$/µL) and percentage of lymphocytes (%).

## Statistical analyses

Data pre-processing, statistical analysis, and plot generation for all datasets was carried out using R (R 4.0 +/RStudio 2022.12.0 +/Bioconductor 3.16+)[59,60] as detailed below. All figures were assembled in Adobe Illustrator v25.1 with select graphical elements from Biorender.

## Data visualization

Comparisons of data distributions between different groups were shown by sina plots using ggplot2 (v3.4.4) and the *geom_sina()* function from the ggforce (v0.4.1) R packages[61,62]. For sina plots, all points were jittered horizontally by local density with overlayed boxes representing medians and interquartile ranges. For comparison of continuous data, XY scatterplots were generated using ggplot2 (v3.4.4)[62]. Points were colored by local density using a custom function. Heatmaps were generated using the ComplexHeatmap (v2.16.0), tidyheatmap (v1.8.1) and ConsensusClusterPlus (v1.64.0) R packages.

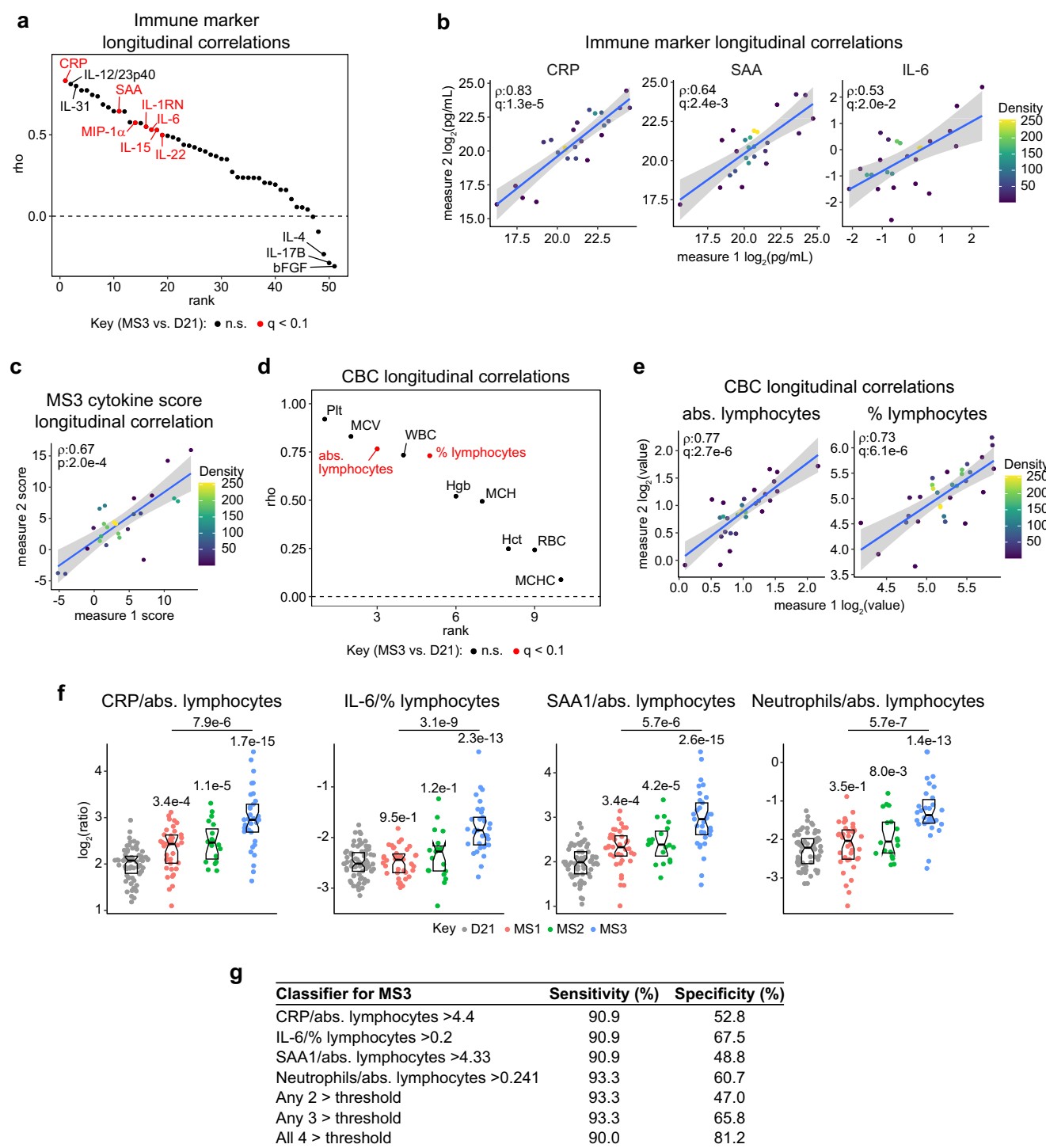

**Fig. 8 | Temporal stability and clinical detection of molecular subtypes of Down syndrome. a** Rank plot showing ranked Spearman correlations of immune markers between longitudinal visits in individuals with T21 ($n = 25$). Red indicates markers altered in MS3. **b** Scatter plots depicting specific examples from (**a**). *rho* and q-values (Benjamini–Hochberg adjusted *p*-values) for Spearman correlation are denoted. Points are colored by density; blue lines represent the fitted values from linear regressions, with 95% confidence intervals in grey. **c** Scatter plot depicting correlation of MS3 cytokine score between longitudinal visits in individuals with T21. *rho* and q-values (Benjamini–Hochberg adjusted *p*-values) for Spearman correlation are denoted. Points are colored by density; blue lines represent the fitted values from linear regressions, with 95% confidence intervals in grey. **d** Rank plot showing ranked Spearman correlations for CBC parameters between longitudinal visits in individuals with T21 ($n = 31$). Red indicates parameters altered in MS3. **e** Scatter plots depicting specific correlations from (**d**). *rho* and q-values (Benjamini–Hochberg adjusted *p*-values) for Spearman correlation are denoted. Points are colored by density; blue

lines represent the fitted values from linear regressions, with 95% confidence intervals in grey. **f** Sina plots showing ratios for CRP to absolute lymphocytes, IL-6 to percent lymphocytes, SAA1 to absolute lymphocytes and neutrophils to absolute lymphocytes in MS1 ($n = 107$, red), MS2 ($n = 95$, green) and MS3 ($n = 102$, blue) compared to euploid controls (D21, $n = 96$, gray). Units employed when using these ratios are relative abundance from SomaScan analyses for CRP, IL-6, and SAA1; percentage among live cells for neutrophils; absolute counts for absolute lymphocytes; and percentage of WBC for percent lymphocytes. q-values, derived from linear regressions and adjusted using the Benjamini–Hochberg method, are displayed above individual data swarms for comparisons to D21 and above lines for MS3 vs. MS1. Boxes represent interquartile ranges and medians, with notches approximating 95% confidence intervals. **g** Table illustrating how ratios shown in (**f**) can be used to discern MS3 with varying sensitivity and specificity. Cutoff values are based on 90th percentile value from euploid controls.

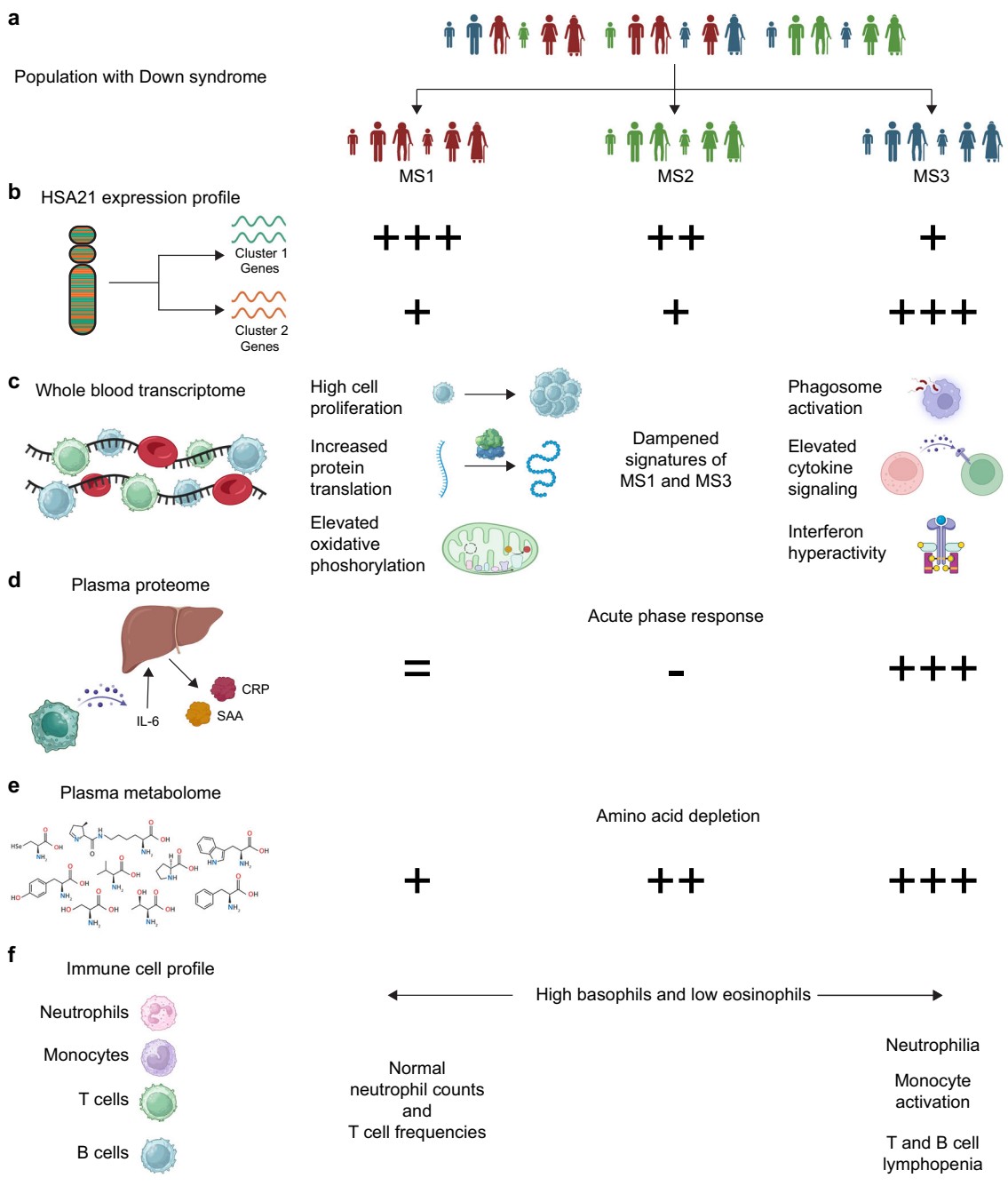

**Fig. 9 | Graphical summary of molecular subtypes of Down syndrome.**
**a** Individuals with trisomy 21 (T21) can be grouped into three distinct molecular subtypes (MS) based on expression of chromosome 21 (HSA21) genes. **b** HSA21 genes are expressed in two distinct co-expression clusters, HSA21 gene cluster 1 (teal) and 2 (orange). MS1 has the highest expression of HSA21 cluster 1 genes relative to euploid controls (D21, +++), followed by MS2 (++), then MS3 (+). MS3 has the highest expression of HSA21 cluster 2 genes relative to D21 (+ + +). MS1 and MS2 overexpress HS21 cluster 2 genes at similar levels (+). **c** The whole blood transcriptome of MS1 is characterized by signatures of high cell proliferation, increased protein translation, and elevated oxidative phosphorylation. MS3 shows the strongest upregulation of signatures indicative of hyperactive immune and inflammatory processes. MS2 has dampened signatures of both MS1 and MS3. **d** Relative to euploid controls, plasma proteomics signatures of the acute phase response are not different for MS1, decreased in MS2, and elevated in MS3. **e** Relative to euploid controls, all subtypes show depletion of plasma amino acids, with increasing severity from MS1 to MS3. **f** All subtypes show elevated basophils and depleted eosinophils, but only MS3 is distinguished by clear neutrophilia concurrent with T and B cell lymphopenia. Panels **a**–**f** created with BioRender.com released under a Creative Commons Attribution-NonCommercial-NoDerivs 4.0 International license.

## Ingenuity pathway analysis (IPA)

Results from DESeq2 analysis of whole-blood transcriptomes (RNA-seq) and linear regressions of plasma proteomics (SomaScan) were analyzed with the use of QIAGEN IPA (version 01-22-01, QIAGEN Inc., https://digitalinsights.qiagen.com/IPA)[63]. Analysis of both RNA-seq and SomaScan datasets specified the reference set as 'User Dataset' and considered only direct relationships. For RNA-seq, we employed cutoffs for both expression fold change (−1.5/1.5) and q-value (0.1), whereas for SomaScan, only a q-value threshold (0.1) was set. Results were exported and visualized using R and RStudio.

## Gene set enrichment analysis (GSEA)

GSEA[64] was carried out in R using the fgsea package (v1.14.0)[65], using Hallmark gene sets[66] and either $log_2$-transformed fold-changes

(for RNA-seq) as previously described[11] or Spearman *rho* values (for correlations) as the ranking metric.

## Spearman correlation and beta regression analysis

Spearman correlation coefficients (*rho*) and *p* values were calculated for all genes in the RNA-seq dataset using the *rcorr ()* function from the Hmisc package (v5.1.1), with Benjamini−Hochberg adjustment of *p* values and an estimated false discovery rate threshold (q) of 0.1. These correlations were subset to individual matrices comparing HSA21 genes to each other and the global transcriptomes. This method was also applied to correlations of HSA21 gene expression vs. relative abundance of plasma SomaScan and MSD analytes and LC-MS metabolites.

## Analysis of whole-blood RNA-seq data

RNA-seq data yield was ~33−103 × $10^6$ raw reads and ~21−69 × $10^6$ final mapped reads per sample. Data quality was assessed using FASTQC (v0.11.5) and FastQ Screen (v0.11.0). Trimming and filtering of low-quality reads was performed using bbduk from BBTools (v37.99)[67] and fastq-mcf from ea-utils (v1.05). Alignment to the human reference genome (GRCh38) was carried out using HISAT2 (v2.1.0)[68] in paired, spliced-alignment mode against a GRCh38 index and Gencode v33 basic annotation GTF, and alignments were sorted and filtered for mapping quality (MAPQ > 10) using Samtools (v1.5)[69]. Gene-level count data were quantified using HTSeq-count (v0.6.1)[70] with the following options (-stranded=reverse −minaqual=10 −type=exon -mode=intersection-nonempty) using a Gencode v33 GTF annotation file. Differential gene expression for individual subtypes of T21 versus both D21 and each other were evaluated using DESeq2 (v1.40.2)[71] with source, sex and age used as covariates. Benjamini−Hochberg adjustment of *p* values was used to adjust for multiple hypotheses, with q < 0.1 (10% FDR) as the threshold for differential expression. Prior to visualization, RPKMs were adjusted for age, sex, and sample source using the *removeBatchEffect()* function from the limma package (v3.56.2)[72].

## Consensus clustering of individuals with DS based on HSA21 gene expression

Clustering of individuals with T21 based on HSA21 gene expression was carried out using the *ConsensusClusterPlus()* function from the ConsensusClusterPlus package (v1.64.0)[73]. RNA-seq data for HSA21 genes were adjusted for differences in source, sex and age using the *removeBatchEffect()* function from the limma package (v3.56.2)[72]. Adjusted RPKMs of HSA21 genes were used to define z-scores among individuals with T21. Adjusted z-scores were used as inputs with 100-fold sub-sampling, Spearman as the distance measure, and agglomerative hierarchical clustering. Examination of the delta area plot and corresponding consensus matrix indicated 3 clusters gave a reasonable compromise between gains in cluster stability and number of clusters.

## Analysis of SomaScan® proteomics data

Normalized data (RFU) in the SomaScan® adat file format was imported to R using the SomaDataIO R package (v3.1.0). Extreme outliers were classified per-karyotype and per-analyte as measurements more than three times the interquartile range below or above the first and third quartiles, respectively (below Q1 − 3*IQR or above Q3 + 3*IQR) and excluded from further analysis. Differential abundance analysis for SomaScan® proteomics was performed using linear regression in R with log2-transformed relative abundance as the outcome/dependent variable, trisomy 21 MS as the predictor/independent variable, and age, sex, and sample source as covariates. Fold changes were calculated between each MS and euploid controls and against each other. Multiple hypothesis correction was performed with the Benjamini−Hochberg method using a false discovery rate (FDR) threshold of 10% (q < 0.1). Prior to visualization or correlation analysis, SomaScan® data were adjusted for age, sex, and sample source using the *removeBatchEffect()* function from the limma package (v3.56.2)[72].

## Analysis of MSD inflammatory marker data

Plasma concentration values (pg/mL) for each of the cytokines and related immune factors measured across multiple MSD assay plates were imported to R, combined, and analytes with >10% of values outside of detection or fit curve range flagged. For each analyte, missing values were replaced with either the minimum (if below fit curve range) or maximum (if above fit curve range) calculated concentration per plate/batch and means of duplicate wells used for subsequent analysis. Extreme outliers were classified per-karyotype and per-analyte as measurements more than three times the interquartile range below or above the first and third quartiles, respectively, and excluded from further analysis. Differential abundance analysis for MSD was performed using linear regression in R with log2-transformed concentration as the outcome/dependent variable, MS as the predictor/independent variable, and age, sex, and sample source as covariates. Fold changes were calculated between each MS and euploid controls and against each other. Multiple hypothesis correction was performed with the Benjamini−Hochberg method using a false discovery rate (FDR) threshold of 10% (q < 0.1). Prior to visualization or correlation analysis, MSD data were adjusted for age, sex, and sample source using the *removeBatchEffect()* function from the limma package (v3.56.2).

## Analysis of mass cytometry data

**Pre-processing.** Normalization and demultiplexing of mass cytometry data were performed using Matlab (version 9.12). Bead-based normalization via polystyrene beads embedded with lanthanides, both within and between batches, followed by bead removal was carried out as previously described using the Matlab-based Normalizer tool[74]. Batched FCS files were demultiplexed using the Matlab-based Single Cell Debarcoder tool[75]. Reference-based normalization of individual samples across batches against the common reference sample was then carried out using the R script *BatchAdjust()*. For the analyses described in this manuscript, CellEngine (CellCarta, version accessed 2022) was used to gate and export per-sample FCS files at four levels: Firstly, CD3 + CD19+ doublets were excluded and remaining cells exported as 'Live' cells; Live cells were then gated for hematopoietic lineage (CD45-positive) non-granulocytic (CD66-low) cells and exported as CD45+CD66low. Lastly, CD45+CD66low cells were gated on CD3-positivity and CD19-positivity and exported as T- and B-cells, respectively. Per-sample FCS files were then subsampled to a maximum of 50,000 events per file for subsequent analysis.

**Unsupervised clustering.** For each of the four levels (live, non-granulocytes, T cells, and B cells), all 388 per-sample FCS files were imported into R as a flowSet object using the *read.flowSet()* function from the flowCore (v2.0.1) R package[76]. Next, a SingleCellExperiment object was constructed from the flowSet object using the *prepData()* function from the CATALYST (v1.12.2) package[77]. Arcsinh transformation was applied to marker expression data with cofactor values ranging from ~0.2 to ~15 to give optimal separation of positive and negative populations for each marker, using the *estParamFlowVS()* function from the flowVS (v1.34.0) R package[78] and based on visual inspection of marker histograms. Quality control and diagnostic plots were examined with the help of functions from CATALYST (v1.12.2) and the tidySingleCellExperiment (v1.3.3) R package. Unsupervised clustering using the FlowSOM algorithm[79] was carried out using the *cluster()* function from CATALYST (v1.12.2), with grid size set to 10 × 10 to give 100 initial clusters and a maxK value of 40 was explored for subsequent meta-clustering using the ConsensusClusterPlus (v1.64.0) algorithm. Examination of delta area and

minimal spanning tree plots indicated that 30–40 meta clusters gave a reasonable compromise between gains in cluster stability and number of clusters for each level. Each clustering level was re-run with multiple random seed values to ensure consistent results.

**Cell type classification.** To aid in assignment of clusters to specific lineages and cell types, the MEM (v3) package (marker enrichment modeling) was used to call positive and negative markers for each cell cluster based on marker expression distributions across clusters. Manual review and comparison to marker expression histograms, as well as minimal spanning tree plots and tSNE plots colored by marker expression, allowed for high-confidence assignment of most clusters to specific cell types. Clusters that were insufficiently distinguishable were merged into their nearest cluster based on the minimal spanning tree. Relative frequencies for each cell type/cluster were calculated for each sample as a percentage of total live cells and as a percentage of cells used for each level of clustering: total CD45+CD66low cells, total T cells, or total B cells.

**Beta regression analysis.** To identify cell clusters for which relative frequencies are associated with trisomy 21 status or subtype, beta regression analysis was carried out using the betareg (v3.1.1-4) R package, with each model using cell type cluster proportions (relative frequency) as the outcome/dependent variable and either T21 status or clinical subgroups as independent/predictor variables, along with adjustment for age and sex, and a logit link function. Extreme outliers were classified per-karyotype and per-cluster as measurements more than three times the interquartile range below or above the first and third quartiles, respectively (below $Q1 - 3*IQR$ or above $Q3 + 3*IQR$) and excluded from beta regression analysis. Correction for multiple comparisons was performed using the Benjamini–Hochberg (FDR) approach. Effect sizes (as fold-change in T21 vs. euploid controls or among T21 subgroups) for each cell type cluster were obtained by exponentiation of beta regression model coefficients. Fold-changes were visualized by overlaying on tSNE plots using ggplot2. For visualization of individual clusters, data points were adjusted for age and sex, using the *adjust()* function from the datawizard (v0.9.0) R package, and visualized as sina plots.

**Analysis of plasma metabolomics and lipidomics data**
Peak intensity data from LC-MS of metabolomics and lipidomics was processed and analyzed using R. Zero values were replaced with random values sampled from 0–0.5x the minimum non-zero value for that metabolite. Data was normalized by applying a scaling factor, which was calculated by dividing the global median intensity value across all metabolites by each sample median intensity. Median normalization was chosen as it is simple to employ, relies on few assumptions, and performs on-par with more complex normalization techniques. Extreme outliers were determined on a per-karyotype and per-analyte basis. Values greater than three times the interquartile range below the first quartile or above the third quartile were considered outliers and are omitted from further analysis. Differential abundance of metabolites was assessed through linear regression using $\log_2$-transformed relative abundance as the outcome/dependent variable, trisomy 21 status as the predictor/independent variable, and age, sex, and sample source as covariates. Multiple hypothesis correction was performed with the Benjamini–Hochberg method using a false discovery rate (FDR) threshold of 10% ($q < 0.1$). Prior to visualization or correlation analysis, metabolite data were adjusted for age, sex, and sample source using the *removeBatchEffect()* function from the limma package (v3.56.2)[72].

**Analysis of co-occurring conditions**
Clinical metadata, including co-occurring conditions and past diagnoses for individuals with trisomy 21 (T21), were analyzed for

conditions present in at least 10 cases in each MS. The prevalence of each condition across molecular subtypes (MS) was compared using Fisher's exact test for pairwise comparisons to identify differences in overrepresentation. These analyses utilized the *tabyl()* and *fisher.test()* functions from the janitor package (v2.2.0).

**Reporting summary**
Further information on research design is available in the Nature Portfolio Reporting Summary linked to this article.

## Data availability
All data needed to evaluate the conclusions in the paper are present in the paper and/or the Supplementary Materials. The demographics and clinical data used in this study have been deposited on the Synapse data sharing platform under accession code syn31488784. The whole blood RNA-seq data used in this study have been deposited on Synapse under accession code syn31488780 and Gene Expression Omnibus under accession code GSE190125. The MSD plasma proteomics data used in this study have been deposited on Synapse under accession code syn31475487. The SomaScan plasma proteomics data used in this study have been deposited on Synapse under accession code syn31488781. The plasma LC-MS metabolomics data used in this study have been deposited on Synapse under accession code syn31488782 and Metabolomics Workbench under accession code ST002200. The immune cell mass cytometry data used in this study have been deposited on Synapse under accession code syn31488783. The entire integrated multidimensional dataset used for this study has been deposited on the Synapse data sharing platform under accession code syn31481952 and the INCLUDE Data Hub (https://portal.includedcc. org/). Please note that the INCLUDE Data Hub does not provide URLs for specific datasets. This is a registered access platform that requires users to create an account and request access to these data files. All data used in this study are also provided in Source data files. Databases used in the generation of these results include KEGG, the Human Molecular Signatures Database (MSigDB, https://www.gsea-msigdb. org/gsea/msigdb/collections.jsp), the Genome Reference Consortium Human Build 38 (GRCh38, https://www.ncbi.nlm.nih.gov/grc/human), and GENCODE Release 33 annotations (https://www.gencodegenes. org/human/release_33.html). Biospecimens are available through the Human Trisome Project Biobank and can be requested online through www.trisome.org. Source data are provided with this paper.

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

## Acknowledgements

We thank all self-advocates with DS and their families for participation in the HTP. We thank Dr. A. D'Alessandro and his team at the Mass Spectrometry Shared Resource for outstanding service in the generation of the metabolomics dataset, Dr. K. Jordan and her team at the Human Immune Monitoring Shared Resource for outstanding service in generation of the immune marker dataset, and Dr. E. Clambey and his team at the Flow Cytometry Shared Resource for outstanding service in generation of the mass cytometry dataset. We are also grateful to the Colorado Translational and Sciences Institute and the Colorado Multiple Institutional Review Board for assistance in all clinical research projects involving the Crnic Institute. Special thanks to M. S. Whitten, the team at the Global Down Syndrome Foundation, Dr. J. Reilly, and Dr. R. Sokol for logistical support at multiple stages of the project. This work was supported by NIH grant R01AI150305 (J.M.E.); NIH grant R01AI145988 (K.D.S.); NIH grant 5UL1TR002535 (J.M.E.); NIH grant P30CA046934 (support of core facilities); the Linda Crnic Institute for Down Syndrome (all authors), the Global Down Syndrome Foundation (all authors), the Anna and John J. Sie Foundation (all authors), GI & Liver Innate Immune Program (J.M.E.), Human Immunology and Immunotherapy Initiative (J.M.E.), University of Colorado School of Medicine (all authors), and Boettcher Foundation (K.D.S.).

## Author contributions

M.G.D. contributed to conceptualization, methodology, investigation, visualization, and writing—original/review and editing (lead author). N.P.E. contributed to conceptualization, methodology (lead analyst), investigation, visualization, and writing—review and editing. K.P.S. contributed to methodology, investigation, and writing—review and editing. E.C.B. contributed to methodology, investigation, and writing—review and editing. H.R.L. contributed to methodology, investigation, and writing—review and editing. P.A. contributed to methodology, investigation, and writing—review and editing. R.E.G. contributed to methodology, investigation, and writing—review and editing. K.A.W. contributed to methodology, investigation, and writing—review and editing. B.E.-E. contributed to methodology, investigation, and writing—review and editing. A.L.R. contributed to methodology, investigation, and writing—review and editing. K.D.S. contributed to conceptualization, methodology, investigation, funding acquisition, supervision, and writing—review and editing. M.D.G. contributed to conceptualization, methodology, investigation, supervision, and writing—review and editing. J.M.E. contributed to conceptualization, methodology, investigation, funding acquisition, supervision, project administration, and writing—original/review and editing.

## Competing interests

J.M.E. has provided consulting services to Eli Lilly and Co., Gilead Sciences Inc., Biohaven Pharmaceuticals, and Perha Pharmaceuticals. All other authors declare that they have no competing interests.
