## [Peer Review File · Nature Communications]

Variegated overexpression of chromosome 21 genes reveals molecular and immune subtypes of Down syndromeREVIEWER COMMENTS

Reviewer #1 (Remarks to the Author):

This is an interesting and very important manuscript that addresses the inter-individual variability of individuals with Down syndrome (DS), also known as trisomy 21 (T21). Its main strength lies in the compelling evidence provided by a relatively large sample from the Human Trisome Project (HTP), comprising 356 research participants with DS and 146 age- and sex-matched euploid controls. The researchers integrated whole blood transcriptomes, plasma proteomes, metabolomes, and immune cell profiles, defining new clusters of TS21 that underlie different pathobiology, likely requiring clinical distinction. This manuscript is particularly timely as clinical trials for molecular therapies in TS21 are underway, and these new targeted personalized approaches need to consider distinct endotypes and phenotypes.

The main weakness of the manuscript is the lack of clarity in some analyses and results. This issue arises from the presentation of numerous datasets and new groups, making the integration of the information challenging for the reader. Despite this, the studies are generally well-described and detailed. This Reviewer has some comments and suggestions to enhance clarity and reinforce the message:

1. In discussing the variegated overexpression of genes on chromosome 21, please specify in all sections of the manuscript that the "clusters" pertain to TS21 genes, not individuals. For instance, instead of "cluster 1" alone, consider using "TS21 genes in cluster 1" because the study also includes clusters of patients. Alternatively, use groups when referring to T21 genes and clusters (instead of molecular subtypes) when referring to T21 patients. This is probably better as the main message is the clustering of T21 patients.

2. Please justify the choice of consensus clustering analysis over other clustering methods. Also, clearly emphasize that this analysis is based on TS21 gene features, not whole transcriptomic signatures.

3. Clarify how the number of clusters was determined, especially as cluster 2 appears intermediate between clusters 1 and 3.

4. As the clustering method is not typical – which makes sense given the uniqueness of TS21 pathobiology- the paper may need an initial workflow/diagram to guide the reader in the two-step method – definition of T21 genes and then consensus clustering analysis-. This probably should be in Figure 1 to improve clarity.

5. Figures 3 to 8: Once clarity is added to results 1 and 2, these subsequent figures are straightforward, depicting significant differences in gene expression, proteins, metabolome, cytokines, and immune phenotypes.

Of note, MS 3 and 1 are clearly different, but 2 appears to be intermediate between the other two – are these really different groups or a spectrum of the same TS21 pathobiology? Please comment on the discussion.

6. I strongly recommend that the authors consider adding a closing figure (or table or diagram) that succinctly summarizes the key features of each of the TS21 clusters of patients (MS1-3). The manuscript presents too much information, and it is hard to integrate the molecular features of the TS21 endotypes and phenotypes by reading the text alone.

7. The abstract, introduction, and discussion are well written. I would only add as a limitation the lack of clinical data integrating the discovered TS21 endotypes and phenotypes. The authors likely have clinical information on the subjects, but this reviewer agrees it is beyond the scope of this already very comprehensive study.

Reviewer #2 (Remarks to the Author):

In NCOMMS-23-55709, Donovan and colleagues present analysis from Human Trisome Project

samples to determine the multiomic variability in Down syndrome (DS). Through a series of transcriptomic, proteomic, metabolomic and blood cell typing experiments, the group shows that 304 people with trisomy 21 can be grouped into 3 molecular subtypes (MS) by variability in Chr21 gene expression. Moreover, individuals within the transcriptome-specified MS types show different proteomic and metabolomic features, further indicating that trisomy 21 (Ts21) can lead to variable phenotypes in individuals. Overall, this important dataset and the accompanying analyses show that assessment of large numbers of DS individuals may elucidate subtype-specific features already present within the biological variability that may be meaningful to their experience of the disorder.

The main critiques of this study are related to technical features of the samples and the comparisons made between them.

Disomic variability: One of the main features of the dataset is the variability in gene expression (those on Chr21 and on other chromosomes) found in individuals with trisomy 21. This is derived from comparison to gene expression records from 92 disomic individuals; the DS plots per individual are then compared to a mean derived from the disomic individuals. This procedure overlooks variability in the disomic population and assumes that the variability in the disomic population is different (less) than that observed in DS population. The first critical revision must be to analyze the control/euploid population to test this critical assumption using unsupervised clustering/PCA analysis for Chr21 and probably the rest of the genome as well.

Technical replicates: As far as the descriptions of transcriptome study allows, it appears that each individual sample was taken once and then sequenced once. Thus, there could be a large amount of variation in gene expression that is due to technical reasons and not to the underlying biology. Ideally, for a cohort like this, multiple samples from the same individual - each sequenced multiple times, are needed to address possible technical artifacts within any given sample. It may be possible to partially address this shortcoming by sequencing a few samples again to show whether the variable genes highlighted in Figure 1 are recovered at the same or different values in each technical repeat.

Multiome correlations: In the study, authors used the transcriptome variability to group DS individuals into one of three molecular subtypes (MS). Then, proteomic and metabolomic values for each MS were compared to one another to test whether the MS grouping is concordant with measures made in the proteomic and metabolomic domains. This order of operation may have forced the comparisons in a biased way. It would be important to test whether the DS proteomes and metabolomes group into 3 clusters in an unbiased way, without prior classification by the authors. In addition, the authors mention that they have curated clinical records from the study individuals, but no information is provided as to whether these people with DS have clinically distinguishable features of the broader DS phenotype, especially corresponding to the MS. Such a correlation between clinical subtype to MS will be impactful, if not fundamental, to the overall validity of the study. It seems important that an advance relating specifically to identifying subtypes of people with DS show that the overall population has clinical subtypes.

Overall, this is a potentially important examination of gene, protein and cell type variability found in the whole blood of people with DS. With necessary revisions and additions to the results, this manuscript will be an important step towards understanding the variability found in the DS population.

Reviewer #3 (Remarks to the Author):

Dear Editor and Authors, I read the manuscript with great interest. The paper addresses a very important issue in understanding DS, namely the molecular correlates of the large phenotypic variability observed in this population.

The authors first find in the analysis of the chromosome 21 transcriptome a 1.5-fold imbalance between DS and controls, as was expected (though debated). However, of particular interest is the large variability of interindividual overexpression of chromosome 21 genes within the sample.

First, the authors describe two gene co-expression clusters with strong positive correlation within the cluster and strong negative correlation between the two clusters.

These two gene clusters are subsequently used to classify the subjects into three molecular subtypes with similar gene transcription. The first subtype with high expression of genes from cluster 1, a second subtype with high expression of genes from cluster 2, and a third subtype with mixed expression of genes from the two clusters.

At this point the authors cascade the differences between these three molecular subtypes in the analysis of the whole blood transcriptome, of the plasma proteome with special focus on inflammatory proteins, of the plasma metabolome, and of leukocyte profiles.

A longitudinal analysis is also performed in a subset of subjects.

I consider the importance of these results very relevant. A clinical reader such as myself, who evaluates many plasma and CBC tests of people with DS every day and finds many oddities in blood values, major differences in leukocyte counts for no apparent reason, or very different responses to the same prescribed medication, in a study such as this one sees the future possibility of having answers and being able to tailor therapeutic intervention.

I think the graphs in the paper are explanatory and the discussion highlights the most important aspects including the limitations of this approach.

I do not consider myself a sufficiently qualified person to evaluate the methods used in this study and I defer to the editor on whether it is appropriate to involve an additional reviewer.

I will add just one small note. There seems to be a mismatch between the people enrolled in the project (356 t21 and 146 controls) and the actual data analyzed and presented (304 t21 and 96 controls). I suggest that the number of subjects actually evaluated (i.e., 304 t21 and 96 controls) should be made more easily visible and if possible also make age, sex and BMI easily available to the reader.

Point by point response to Reviewers.

Manuscript NCOMMS-23-55709

REVIEWER COMMENTS

Reviewer #1 (Remarks to the Author):

This is an interesting and very important manuscript that addresses the inter-individual variability of individuals with Down syndrome (DS), also known as trisomy 21 (T21). Its main strength lies in the compelling evidence provided by a relatively large sample from the Human Trisome Project (HTP), comprising 356 research participants with DS and 146 age- and sex-matched euploid controls. The researchers integrated whole blood transcriptomes, plasma proteomes, metabolomes, and immune cell profiles, defining new clusters of TS21 that underlie different pathobiology, likely requiring clinical distinction. This manuscript is particularly timely as clinical trials for molecular therapies in TS21 are underway, and these new targeted personalized approaches need to consider distinct endotypes and phenotypes.

The main weakness of the manuscript is the lack of clarity in some analyses and results. This issue arises from the presentation of numerous datasets and new groups, making the integration of the information challenging for the reader. Despite this, the studies are generally well-described and detailed. This Reviewer has some comments and suggestions to enhance clarity and reinforce the message:

1. In discussing the variegated overexpression of genes on chromosome 21, please specify in all sections of the manuscript that the "clusters" pertain to TS21 genes, not individuals. For instance, instead of "cluster 1" alone, consider using "TS21 genes in cluster 1" because the study also includes clusters of patients. Alternatively, use groups when referring to T21 genes and clusters (instead of molecular subtypes) when referring to T21 patients. This is probably better as the main message is the clustering of T21 patients.

Response: We appreciate this insightful feedback and understand the potential confusion arising from our terminology, particularly since both genes and groups of individuals with T21 were identified through clustering exercises. To avoid ambiguity, we have adopted the first suggestion from the Reviewer to consistently use 'HSA21 **gene** cluster 1' or 'HSA21 **gene** cluster 2' throughout all pertinent sections of the text and figures when referring to the two sets of co-expressed human chromosome 21 (HSA21) genes. This revision ensures a clear distinction in our manuscript, explicitly differentiating the **gene clusters** from the identified **molecular subtypes** among individuals with T21.

2. Please justify the choice of consensus clustering analysis over other clustering methods. Also, clearly emphasize that this analysis is based on TS21 gene features, not whole transcriptomic signatures.

Response: We appreciate the Reviewer's inquiry regarding our choice of consensus clustering, specifically from the ConsensusClusterPlus package¹, for our analysis. We selected this approach due to its robustness and established reputation in the literature for similar types of analyses where individuals are clustered based on gene expression data¹⁻⁵. Consensus clustering extends the strengths of hierarchical clustering by incorporating an iterative process. This feature is particularly advantageous in our study, as it rigorously assesses the stability and consistency of the clusters over multiple iterations, ensuring that our results are not artifacts of a particular run or initialization. This iterative nature is key to handling the complexity and variability of the T21 gene expression data. By repeatedly applying hierarchical clustering across a range of k-values, it provides a comprehensive assessment of cluster number and stability. While there are other clustering methods available, the iterative refinement offered by consensus clustering sets it apart, making it particularly suitable for our analysis which focuses on chromosome 21 gene expression.

We are thankful for the suggestion to more explicitly clarify that our consensus clustering analysis focuses on chromosome 21 gene expression, rather than encompassing the entire transcriptome. In response to your feedback, we have carefully reviewed the relevant sections, particularly in the

corresponding Results and Methods sections, and have revised the language to ensure that the specific focus on chromosome 21 gene expression is unmistakably clear. Additionally, we believe the new illustration outlining our two-part analysis, which we created in response to the Reviewer's comment #4 below, will also make this clearer for the readers.

3. Clarify how the number of clusters was determined, especially as cluster 2 appears intermediate between clusters 1 and 3.

Response: Thank you for your inquiry regarding our determination of the number of clusters among individuals with T21 (i.e., Molecular Subtypes, MS), particularly in reference to MS2, which appears intermediate between MS1 and MS3. In our analysis, we employed the consensus clustering method from the ConsensusClusterPlus package¹, which produces a delta area plot as a standard output. The delta area plot is a crucial tool in determining the optimal number of clusters, as it visualizes the relative change in area under the cumulative distribution function (CDF) curve for each number of clusters (k). A larger relative change indicates a more distinct clustering solution. In our case, the plot showed that while a viable clustering solution is observable at k=2, the delineation of clusters becomes more defined and robust at k=3 (see manuscript **Supplementary Fig. 2b**). Advancing beyond k=3 led to only marginal improvements in the relative change, indicating that k=3 was the optimal choice for our study, effectively balancing statistical robustness and interpretability (**Supplementary Fig. 2b**). To clarify for readers how the number of clusters was determined, we have included a description of this thought process in the relevant sections in both the Results and Methods sections. Furthermore, the intermediate nature of MS2, situated between MS1 and MS3, is indeed a noteworthy observation. Our comparison of HSA21 gene cluster polygenic expression scores suggests that while this intermediate phenotype may be true for HSA21 **cluster 1** genes, this may not be the case for HSA21 **cluster 2** genes (**Fig. 2f**). Moreover, MS2 showed multiomic features beyond HSA21 gene expression that were unique from the other subtypes.

4. As the clustering method is not typical – which makes sense given the uniqueness of TS21 pathobiology- the paper may need an initial workflow/diagram to guide the reader in the two-step method – definition of T21 genes and then consensus clustering analysis-. This probably should be in Figure 1 to improve clarity.

Response: We thank the Reviewer for this valuable suggestion. We agree that incorporating a workflow diagram would significantly clarify our analysis workflow. While we appreciate the recommendation to place this diagram in **Fig. 1**, we found that later in the manuscript is a more suitable location, when the results of **Fig. 2** are described, as it is in this instance in the paper where we describe the consensus clustering analyses and downstream exercises. To this end, we have now included an illustration of our approach in **Supplementary Fig. 2a** and reference to it in the relevant

Reviewer Fig. 1 | Overview of analysis methods used in this manuscript. Whole blood RNA-seq data from individuals with trisomy 21 (T21) were analyzed to identify co-expression clusters of chromosome 21 (HSA21) genes and to classify molecular subtypes within the T21 group based on HSA21 gene expression. Unsupervised hierarchical clustering, utilizing Spearman correlations of RPKMs, defined the HSA21 gene clusters. The molecular subtypes were delineated through consensus clustering of z-scored RPKM data, calculated from the mean and standard deviation exclusively within the T21 cohort's HSA21 expression data.

section of the text. This addition will provide readers with a clear visual guide through the methodology employed in our study. Please note that due to this addition, the labeling of subsequent panels in **Supplementary Fig. 2** has been sequentially incremented to accommodate the new figure panel. This new figure panel is also shared here with Reviewers as **Reviewer Fig. 1**.

5. Figures 3 to 8: Once clarity is added to results 1 and 2, these subsequent figures are straightforward, depicting significant differences in gene expression, proteins, metabolome, cytokines, and immune phenotypes. Of note, MS 3 and 1 are clearly different, but 2 appears to be intermediate between the other two – are these really different groups or a spectrum of the same TS21 pathobiology? Please comment on the discussion.

Response: We appreciate the Reviewer's observation regarding the potential intermediate nature of MS2 and the query of whether the molecular subtypes represent distinct groups or a spectrum within T21 pathobiology. Our findings suggest a nuanced phenotype for MS2, where certain molecular features seem to be intermediate between MS1 and MS3, while other features are clearly unique. For example, MS2 exhibits intermediate expression of HSA21 cluster 1 genes (manuscript **Fig. 2f**), activation of phagosome formation in the whole blood transcriptome (manuscript **Fig. 3b**), and inhibition of intrinsic prothrombin activation (manuscript **Fig. 4b**). Conversely, MS2 shows an expression profile similar to that of MS1 for HSA21 cluster 2 genes (manuscript **Fig. 2f**) and is the only subtype to display proteomic changes indicative of inhibition of the acute phase response and IL-15 production (manuscript **Fig. 4b**). Given these nuances, we remain cautious in definitively categorizing these subtypes as entirely separate groups or as points along a continuum. However, we do acknowledge that these subtypes are not genetically defined and are instead based on mRNA expression patterns, which may exhibit a degree of fluidity. We partially addressed the stability aspect of these subtypes in the manuscript (manuscript **Fig. 8**). Here we show how values for certain cytokines and complete blood count parameters maintain stability over a long period of time (>1 year). However, we remain open to the possibility that individuals may exhibit a gradient of features, potentially navigating across subtypes over time or under different physiological conditions. Following Reviewer's guidance, we have expanded upon this concept in the Discussion section of our manuscript.

6. I strongly recommend that the authors consider adding a closing figure (or table or diagram) that succinctly summarizes the key features of each of the TS21 clusters of patients (MS1-3). The manuscript presents too much information, and it is hard to integrate the molecular features of the TS21 endotypes and phenotypes by reading the text alone.

Response: Thank you for the valuable suggestion to include a comprehensive figure that clearly outlines the key attributes of each T21 molecular subtype (MS1-3). We agree that a visual summary will greatly enhance the manuscript by providing an integrated view of the molecular subtypes (MS) with respect to their HSA21 gene expression profiles and the distinctive features of their whole-blood transcriptomes, plasma proteomes/metabolomes, and immune cell populations. In response, we have added a new **Fig. 9**, which concisely illustrates the unique molecular signatures and distinguishing multiomic aspects of MS1-3. This addition facilitates a more intuitive understanding of our findings. This new figure panel is also shared here with Reviewers as **Reviewer Fig. 2**:

Reviewer Fig. 2 | Graphical summary of molecular subtypes of Down syndrome.

a Individuals with trisomy 21 (T21) can be grouped into three distinct molecular subtypes (MS) based on expression of chromosome 21 (HSA21) genes. **b** HSA21 genes are expressed in two distinct co-expression clusters, HSA21 gene cluster 1 and 2. MS1 has the highest expression of HSA21 cluster 1 genes relative to euploid controls (D21, +++), followed by MS2 (++), then MS3 (+). MS3 has the highest expression of HSA21 cluster 2 genes relative to D21 (+++). MS1 and MS2 overexpress HSA21 cluster 2 genes at similar levels (+). **c** The whole blood transcriptome of MS1 is characterized by signatures of high cell proliferation, increased protein translation, and elevated oxidative phosphorylation. MS3 shows the strongest upregulation of signatures indicative of hyperactive immune and inflammatory processes. MS2 has dampened signatures of both MS1 and MS3. **d** Relative to euploid controls, plasma proteomics signatures of the acute phase response are not different for MS1, decreased in MS2, and elevated in MS3. **e** Relative to euploid controls, all subtypes show depletion of plasma amino acids, with increasing severity from MS1 to MS3. **f** All subtypes show elevated basophils and depleted eosinophils, but only MS3 is distinguished by clear neutrophilia concurrent with T and B cell lymphopenia.

7. The abstract, introduction, and discussion are well written. I would only add as a limitation the lack of clinical data integrating the discovered TS21 endotypes and phenotypes. The authors likely have clinical information on the subjects, but this reviewer agrees it is beyond the scope of this already very comprehensive study.

Response: We appreciate the Reviewer's commendation of the Abstract, Introduction, and Discussion sections of our manuscript, and we understand the noted limitation regarding the integration of clinical data in our comparison of the molecular subtypes. Despite our best efforts, our analysis revealed only modest trends in the overrepresentation of certain co-occurring conditions within specific subtypes, which did not achieve statistical significance (see **Reviewer Fig. 3**). This outcome is likely due to diminished statistical power resulting from the stratification of our T21 cohort into three distinct groups, consequently reducing the number of cases available for each condition. Given this constraint, we are hesitant to publish these preliminary observations without a larger dataset that could lend to statistical significance. Our hope is that this study will pave the road towards a deeper understanding of the heterogeneity within T21 pathobiology and will galvanize further research efforts aimed at elucidating the differences in the risk of co-occurring conditions among individuals with DS.

Reviewer Fig. 3 | Molecular subtypes of Down syndrome show trends toward differential prevalence of co-occurring conditions. Pairwise comparisons between trisomy 21 (T21) molecular subtypes (MS) on the overrepresentation of cases vs. controls for co-occurring conditions associated with Down syndrome (DS). Forest plots show odds ratios (boxes) with 95% confidence intervals (whiskers).

Reviewer #2 (Remarks to the Author):

In NCOMMS-23-55709, Donovan and colleagues present analysis from Human Trisome Project samples to determine the multiomic variability in Down syndrome (DS). Through a series of transcriptomic, proteomic, metabolomic and blood cell typing experiments, the group shows that 304 people with trisomy 21 can be grouped into 3 molecular subtypes (MS) by variability in Chr21 gene expression. Moreover, individuals within the transcriptome-specified MS types show different proteomic and metabolomic features, further indicating that trisomy 21 (Ts21) can lead to variable phenotypes in individuals. Overall, this important dataset and the accompanying analyses show that assessment of large numbers of DS individuals may elucidate subtype-specific features already present within the biological variability that may be meaningful to their experience of the disorder.

The main critiques of this study are related to technical features of the samples and the comparisons made between them.

Disomic variability: One of the main features of the dataset is the variability in gene expression (those on Chr21 and on other chromosomes) found in individuals with trisomy 21. This is derived from comparison to gene expression records from 92 disomic individuals; the DS plots per individual are then compared to a mean derived from the disomic individuals. This procedure overlooks variability in the disomic population and assumes that the variability in the disomic population is different (less) than that observed in DS population. The first critical revision must be to analyze the control/euploid population to test this critical assumption using unsupervised clustering/PCA analysis for Chr21 and probably the rest of the genome as well.

Response: We appreciate the opportunity to clarify a critical aspect of our methodology that seems to have led to some misunderstanding. It is important to highlight that the T21 subtypes were delineated by a consensus clustering algorithm employed by ConsensusClusterPlus¹, which was applied to the z-scores of HSA21 gene expression RPKMs. **Crucially, these z-scores were calculated based on the mean and standard deviation specific to the T21 cohort alone, without reference to the D21 (euploid) controls.** Although, for context, Fig. 1a in the manuscript presents gene expression changes relative to euploid individuals, the actual clustering process was confined entirely to the T21 cohort. This approach ensures that our subtype definition is completely agnostic to the variability present in the D21 population. We made no assumptions regarding the degree of variability in the D21 group, and therefore, such variability - or the lack thereof - has no bearing on the outcome of the clustering exercise.

However, inspired by this Reviewer's request, we conducted consensus clustering for HSA21 gene expression in the D21 control cohort. While this analysis does reveal some variability within the D21 group, the separation of clusters is less distinct than that observed in the T21 samples. This is illustrated by the consensus matrices for clustering solutions with $k = 2$ and $k = 3$, which show some level of clustering but with less pronounced separation than what was observed for $k = 3$ in the T21 group (**Reviewer Fig. 4**). Although these findings provide insights into the variability within the D21 population, they do not alter the fundamental interpretation of our results as presented in our manuscript. Given our focus on the study of DS, we share this clustering exercise for the D21 cohort in the response to Reviewer's only.

Reviewer Fig. 4 | Consensus clustering of HSA21 gene expression in euploid controls compared to individuals with trisomy 21. a Consensus matrices showing clustering of HSA21 genes euploid control individuals based on a $k = 2$ (left) and $k = 3$ (right) clustering solution. b Consensus matrix showing clustering of HSA21 genes in individuals with T21 based on a $k = 3$ clustering solution.

Technical replicates: As far as the descriptions of transcriptome study allows, it appears that each individual sample was taken once and then sequenced once. Thus, there could be a large amount of variation in gene expression that is due to technical reasons and not to the underlying biology. Ideally, for a cohort like this, multiple samples from the same individual - each sequenced multiple times, are needed to address possible technical artifacts within any given sample. It may be possible to partially address this shortcoming by sequencing a few samples again to show whether the variable genes highlighted in Figure 1 are recovered at the same or different values in each technical repeat.

Response: We acknowledge the Reviewer's concern regarding the need to mitigate potential technical variations in gene expression analyses. In whole blood RNA-seq studies, it is not uncommon to rely on a single sequencing run per sample due to the high sensitivity and specificity of modern sequencing technologies^{6,7} and the fact that there is no experimental procedure between sample extraction and RNA preservation. The whole blood transcriptome data presented in the manuscript was generated from RNA extracted from PAXgene RNA tubes, which immediately preserve the RNA in the bloodstream at the time of the phlebotomy. Unlike RNA extracted from cell cultures or animal models, there is no 'experimental protocol' that could lead to operator-induced variation in gene expression. In this regard, the procedure is similar to a clinical laboratory test (e.g., liver enzymes, cell blood count with differential) commonly used in the clinical setting. However, despite this standard approach, to ease the Reviewer's concern, we undertook additional sequencing efforts on two separate whole blood samples from a subset of individuals (n = 12), with specimens collected approximately one week apart. Collecting two identical samples during the same phlebotomy would have required strong scientific justification and regulatory approval from the Institutional Review Board. To directly assess the consistency of gene expression measurements a week apart, we compared RPKMs for all genes detected in the RNA-seq across these two distinct time points for all 12 individuals (**Reviewer Fig. 5**). This exercise demonstrates a high degree of concordance between the two time points reinforcing the reliability of our gene expression data (rho values always higher than 0.9). These results also point to the stability of the gene expression signatures for individuals within a short period of time.

Reviewer Fig. 5 | Temporal stability of RNA-seq measurements. Scatter plots comparing RPKMs for all genes detected in RNA-seq from two different samples from the same individuals (n = 12) obtained approximately one week apart. Points are colored by density; blue lines represent linear model fit with 95% confidence intervals in grey. Pearson correlation values are displayed.

Multiome correlations: In the study, authors used the transcriptome variability to group DS individuals into one of three molecular subtypes (MS). Then, proteomic and metabolomic values for each MS were compared to one another to test whether the MS grouping is concordant with measures made in the proteomic and metabolomic domains. This order of operation may have forced the comparisons in a biased way. It would be important to test whether the DS proteomes and metabolomes group into 3 clusters in an unbiased way, without prior classification by the authors. In addition, the authors mention that they have curated clinical records from the study individuals, but no information is provided as to

whether these people with DS have clinically distinguishable features of the broader DS phenotype, especially corresponding to the MS. Such a correlation between clinical subtype to MS will be impactful, if not fundamental, to the overall validity of the study. It seems important that an advance relating specifically to identifying subtypes of people with DS show that the overall population has clinical subtypes.

Response: We appreciate the Reviewer's comment about the importance of exploring other modes of stratification based on other available data, such as proteome or metabolome. We would like to point out that our methodology was driven by the novel observation that HSA21 overexpression is highly variable among individuals with T21, which we believe may hold key insights for DS pathobiology. This observation led to the hypothesis that HSA21 expression variability might be indicative of wider molecular heterogeneity within the DS population and potentially mirrored in proteomic and metabolomic signatures. By classifying individuals into molecular subtypes based on HSA21 expression data, our goal was to assess if these subtypes exhibited distinct proteomic and metabolomic profiles, thus providing a more integrated understanding of molecular heterogeneity in DS.

We agree with the Reviewer that other datasets (e.g., proteomics, metabolomics) could also be employed to identify subgroups of individuals with DS, which could indeed yield valuable insights. However, it is crucial to acknowledge the broad spectrum of additional features available for clustering, including immune cell profiles, clinical data, socio-economic status, dietary and physical activity habits, among many others. This highlights the vast landscape of potential clustering dimensions, underscoring the importance of selecting a focus that is both practical and impactful for the confines of a single study. To illustrate the value of our approach, consider the example of BRCA1-positive women who have an increased risk of breast cancer mortality. This specific genetic classifier offers crucial insights about breast cancer risk. Yet, it is recognized that other non-genetic factors such as lack of medical insurance or nulliparity also independently affect breast cancer mortality. Similarly, the molecular subtypes identified based on HSA21 gene expression offer a precise framework to explore the molecular heterogeneity of DS within the scope of this study, without negating the possibility that other factors can also stratify the population with DS, such as proteomic or metabolomic signatures, genetic and epigenetic mechanisms⁸⁻¹², sex¹³, race and ethnicity¹⁴⁻¹⁷, socio-economic status¹⁸⁻²⁰, diet²¹, and cultural background²².

In response to the Reviewer's comment regarding the lack clinical features associated with each molecular subtype (MS), we would like to note that this limitation was acknowledged in the Discussion. As explained above in response to one of Reviewers #1 comments, our exploration of co-occurring conditions within the molecular subtypes did uncover some intriguing trends. However, these findings did not achieve statistical significance (see above, **Reviewer Fig. 3** see above). This outcome is likely due to diminished statistical power resulting from the stratification of our T21 cohort into three distinct groups, consequently reducing the number of cases available for each condition. Given this constraint, we are hesitant to publish these preliminary observations without a larger dataset that could lead to statistically significant findings. Our hope is that this study will pave the road towards a deeper understanding of the heterogeneity within T21 pathobiology and will galvanize further research efforts aimed at elucidating the differences in the risk of co-occurring conditions among individuals with DS through larger cohort studies and large-scale multiomics datasets.

Overall, this is a potentially important examination of gene, protein and cell type variability found in the whole blood of people with DS. With necessary revisions and additions to the results, this manuscript will be an important step towards understanding the variability found in the DS population.

Response: We agree with this comment and appreciate the acknowledgment of the potential importance of our study in examining the molecular heterogeneity of individuals with Down syndrome. Thank you for your constructive feedback.

Reviewer #3 (Remarks to the Author):

Dear Editor and Authors, I read the manuscript with great interest. The paper addresses a very important issue in understanding DS, namely the molecular correlates of the large phenotypic variability observed in this population.

The authors first find in the analysis of the chromosome 21 transcriptome a 1.5-fold imbalance between DS and controls, as was expected (though debated). However, of particular interest is the large variability of interindividual overexpression of chromosome 21 genes within the sample.

First, the authors describe two gene co-expression clusters with strong positive correlation within the cluster and strong negative correlation between the two clusters.

These two gene clusters are subsequently used to classify the subjects into three molecular subtypes with similar gene transcription. The first subtype with high expression of genes from cluster 1, a second subtype with high expression of genes from cluster 2, and a third subtype with mixed expression of genes from the two clusters.

At this point the authors cascade the differences between these three molecular subtypes in the analysis of the whole blood transcriptome, of the plasma proteome with special focus on inflammatory proteins, of the plasma metabolome, and of leukocyte profiles. A longitudinal analysis is also performed in a subset of subjects.

I consider the importance of these results very relevant. A clinical reader such as myself, who evaluates many plasma and CBC tests of people with DS every day and finds many oddities in blood values, major differences in leukocyte counts for no apparent reason, or very different responses to the same prescribed medication, in a study such as this one sees the future possibility of having answers and being able to tailor therapeutic intervention.

I think the graphs in the paper are explanatory and the discussion highlights the most important aspects including the limitations of this approach.

I do not consider myself a sufficiently qualified person to evaluate the methods used in this study and I defer to the editor on whether it is appropriate to involve an additional reviewer.

I will add just one small note. There seems to be a mismatch between the people enrolled in the project (356 t21 and 146 controls) and the actual data analyzed and presented (304 t21 and 96 controls). I suggest that the number of subjects actually evaluated (i.e., 304 t21 and 96 controls) should be made more easily visible and if possible also make age, sex and BMI easily available to the reader.

Response: We appreciate the Reviewer's kind words and thoughtful engagement with our manuscript. Your recognition of the importance of our findings in understanding the molecular underpinnings of phenotypic variability in Down syndrome is encouraging. We are especially grateful for your perspective as a clinical reader, which underscores the potential clinical implications and utility of our research.

Regarding the Reviewer's observation about the discrepancy between the number of total participants enrolled and those included in specific analysis, this is explained by the fact that, within a larger cohort of **total** participants involved in this report, *mostly overlapping* subsets of this cohort were employed for specific assays (e.g., transcriptome, proteome, metabolome, cell blood counts) from the exact same blood sample to enable the cross-omics analyses described in our manuscript. However, as illustrated in **Reviewer Fig. 6**, the overlap is not complete, with some participants/samples contributing additional data points to some of the datasets. Thus, the total size of the cohort analyzed in our manuscript is greater (356 T21 and 146 controls) than the subset used for the transcriptome study (304 T21 and 96 controls). In the manuscript, we sought to provide clarity by specifying the sample numbers within the legends of each figure panel. To further enhance transparency in our revised submission, we have made it a point to explicitly state the number of individuals contributing to each dataset whenever we introduce a new data type in the results section.

Reviewer Fig. 6 | Overlaps between matched blood samples across multiomic datasets. a Upset plot showing overlaps between matched blood samples across datasets.

To address the comment requesting clinical variables be accessible to readers, we have added a sheet to **Supplementary File 2**, which includes the karyotype, molecular subtype, sex, age, **and BMI** for all participants. In addition, for the new submission, we have included **Fig. 2d**, which plots the **BMI** distributions across the groups with accompanying statistics from Wilcoxon rank-sum tests. These are adjacent to **Fig. 2c**, which does the same exercise for age. Please note, due to the inclusion of this new figure panel, the labeling of subsequent panels in **Fig. 2** has been sequentially incremented to accommodate the new figure panel.

References.

- 1 Wilkerson, M. D. & Hayes, D. N. ConsensusClusterPlus: a class discovery tool with confidence assessments and item tracking. *Bioinformatics* **26**, 1572-1573 (2010). <https://doi.org/10.1093/bioinformatics/btq170>
- 2 Qiu, C. *et al.* Identification of Molecular Subtypes and a Prognostic Signature Based on Inflammation-Related Genes in Colon Adenocarcinoma. *Frontiers in immunology* **12**, 769685 (2021). <https://doi.org/10.3389/fimmu.2021.769685>
- 3 Tan, Z. *et al.* Integrating Bulk and Single-Cell RNA Sequencing Reveals Heterogeneity, Tumor Microenvironment, and Immunotherapeutic Efficacy Based on Sialylation-Related Genes in Bladder Cancer. *J Inflamm Res* **16**, 3399-3417 (2023). <https://doi.org/10.2147/jir.S418433>
- 4 Yang, Z., Wang, G., Luo, N., Tsang, C. K. & Huang, L. Consensus clustering of gene expression profiles in peripheral blood of acute ischemic stroke patients. *Front Neurol* **13**, 937501 (2022). <https://doi.org/10.3389/fneur.2022.937501>
- 5 Yang, C., Wang, G., Zhan, W., Wang, Y. & Feng, J. The identification of metabolism-related subtypes and potential treatments for idiopathic pulmonary fibrosis. *Front Pharmacol* **14**, 1173961 (2023). <https://doi.org/10.3389/fphar.2023.1173961>
- 6 Marioni, J. C., Mason, C. E., Mane, S. M., Stephens, M. & Gilad, Y. RNA-seq: an assessment of technical reproducibility and comparison with gene expression arrays. *Genome research* **18**, 1509-1517 (2008). <https://doi.org/10.1101/gr.079558.108>
- 7 Liu, Y., Zhou, J. & White, K. P. RNA-seq differential expression studies: more sequence or more replication? *Bioinformatics* **30**, 301-304 (2014). <https://doi.org/10.1093/bioinformatics/btt688>
- 8 Prandini, P. *et al.* Natural gene-expression variation in Down syndrome modulates the outcome of gene-dosage imbalance. *American journal of human genetics* **81**, 252-263 (2007). <https://doi.org/10.1086/519248>

- 9 Ramachandran, D. *et al.* Contribution of copy-number variation to Down syndrome-associated atrioventricular septal defects. *Genetics in medicine : official journal of the American College of Medical Genetics* **17**, 554-560 (2015). <https://doi.org/10.1038/gim.2014.144>
- 10 Brown, A. L. *et al.* Inherited genetic susceptibility to acute lymphoblastic leukemia in Down syndrome. *Blood* **134**, 1227-1237 (2019). <https://doi.org/10.1182/blood.2018890764>
- 11 El Hajj, N. *et al.* Epigenetic dysregulation in the developing Down syndrome cortex. *Epigenetics* **11**, 563-578 (2016). <https://doi.org/10.1080/15592294.2016.1192736>
- 12 Yu, Y. E. *et al.* Genetic and epigenetic pathways in Down syndrome: Insights to the brain and immune system from humans and mouse models. *Prog Brain Res* **251**, 1-28 (2020). <https://doi.org/10.1016/bs.pbr.2019.09.002>
- 13 Andrews, E. J., Martini, A. C. & Head, E. Exploring the role of sex differences in Alzheimer's disease pathogenesis in Down syndrome. *Frontiers in neuroscience* **16**, 954999 (2022). <https://doi.org/10.3389/fnins.2022.954999>
- 14 Iulita, M. F. *et al.* Association of Alzheimer Disease With Life Expectancy in People With Down Syndrome. *JAMA Netw Open* **5**, e2212910 (2022). <https://doi.org/10.1001/jamanetworkopen.2022.12910>
- 15 Chaiken, S. R., Susich, M., Doshi, U., Packer, C.H., Garg, B., Caughey, A.B. Down Syndrome Trends by Race/Ethnicity in the United States from 2012-2018. *American Journal of Obstetrics and Gynecology*, S481-482 (2022). <https://doi.org/https://doi.org/10.1016/j.ajog.2021.11.797>
- 16 Khoshnood, B. *et al.* Ethnic differences in the impact of advanced maternal age on birth prevalence of Down syndrome. *Am J Public Health* **90**, 1778-1781 (2000). <https://doi.org/10.2105/ajph.90.11.1778>
- 17 Kruszka, P. *et al.* Down syndrome in diverse populations. *American journal of medical genetics. Part A* **173**, 42-53 (2017). <https://doi.org/10.1002/ajmg.a.38043>
- 18 Budd, J. L., Draper, E. S., Lotto, R. R., Berry, L. E. & Smith, L. K. Socioeconomic inequalities in pregnancy outcome associated with Down syndrome: a population-based study. *Arch Dis Child Fetal Neonatal Ed* **100**, F400-404 (2015). <https://doi.org/10.1136/archdischild-2014-306985>
- 19 Hunter, J. E. *et al.* The association of low socioeconomic status and the risk of having a child with Down syndrome: a report from the National Down Syndrome Project. *Genetics in medicine : official journal of the American College of Medical Genetics* **15**, 698-705 (2013). <https://doi.org/10.1038/gim.2013.34>
- 20 Kucik, J. E. *et al.* Trends in survival among children with Down syndrome in 10 regions of the United States. *Pediatrics* **131**, e27-36 (2013). <https://doi.org/10.1542/peds.2012-1616>
- 21 Roccatello, G. *et al.* Eating and Lifestyle Habits in Youth With Down Syndrome Attending a Care Program: An Exploratory Lesson for Future Improvements. *Front Nutr* **8**, 641112 (2021). <https://doi.org/10.3389/fnut.2021.641112>
- 22 Gatford, A. Down's syndrome: experiences of mothers from different cultures. *Br J Nurs* **10**, 1193-1199 (2001). <https://doi.org/10.12968/bjon.2001.10.18.9940>

REVIEWERS' COMMENTS

Reviewer #1 (Remarks to the Author):

My comments have been addressed. The revised paper has a clearer presentation of methods and results. No additional concerns.

Reviewer #2 (Remarks to the Author):

I appreciate the detailed response to the critiques provided by the authors. I agree that describing possible Ts21 individual variation via blood sample gene expression is a useful way to start understanding the variability of clinical findings in the population. As such, the information in this report is important to publish for the field. However, the inability to correlate any of the three expression-related groups to clinical outcomes weakens the impact of the report. It is not quite clear why samples are sufficient for the clustering and not the clinical conditions. It may well be that the source of the sequencing samples (blood) does not report salient features present in other tissues that would better align with clinical findings. Furthermore, since the authors appear reluctant to demonstrate this discontinuity (by not reporting these preliminary clinical findings), this may lead to misunderstandings of the power and relevance of this report by readers.

In general, this paper provides new information and methods for possible future examination of the genotype/phenotype relationship in Down syndrome. However, it is critical that the caveats noted above are presented in the manuscript.

Response to Reviewer's Comments:

Reviewer #1 (Remarks to the Author):

My comments have been addressed. The revised paper has a clearer presentation of methods and results. No additional concerns.

Response: We thank this Reviewer for the feedback and for acknowledging the improvements made to the manuscript. We are pleased that the revisions have addressed their concerns and enhanced the clarity of our presentation. We appreciate the insightful comments which have undoubtedly helped strengthen our paper.

Reviewer #2 (Remarks to the Author):

I appreciate the detailed response to the critiques provided by the authors. I agree that describing possible Ts21 individual variation via blood sample gene expression is a useful way to start understanding the variability of clinical findings in the population. As such, the information in this report is important to publish for the field. However, the inability to correlate any of the three expression-related groups to clinical outcomes weakens the impact of the report. It is not quite clear why samples are sufficient for the clustering and not the clinical conditions. It may well be that the source of the sequencing samples (blood) does not report salient features present in other tissues that would better align with clinical findings. Furthermore, since the authors appear reluctant to demonstrate this discontinuity (by not reporting these preliminary clinical findings), this may lead to misunderstandings of the power and relevance of this report by readers.

In general, this paper provides new information and methods for possible future examination of the genotype/phenotype relationship in Down syndrome. However, it is critical that the caveats noted above are presented in the manuscript.

Response: We thank the Reviewer for their additional comments and concerns regarding the limitations related to the correlation of the molecular subtypes (i.e., expression-related groups) with clinical outcomes. In response to this concern, and based on editorial guidance, we have included in the revised manuscript the results from pairwise comparisons between T21 molecular subtypes on the overrepresentation of cases vs. controls for common co-occurring conditions associated with DS. This was originally presented as Reviewer Figure 3 in our previous response to Reviewers. While these results did not reach statistical significance after multiple hypotheses correction, they provide valuable insights into the challenges of correlating groups defined from RNA-seq measurements with clinical manifestations. We acknowledge that blood may not be the ideal biomarker for identifying all clinical differences due to its inability to fully represent the molecular activities of other tissues more directly involved in the conditions associated with DS. In response to the comment as to why samples are sufficient for clustering based on RNA-seq measurements, but not for identifying differences in clinical conditions, we acknowledge that other tissue-specific

biosignatures (e.g. brain, gut, lung, liver) may be more powerful to define associations to clinical variables. Nevertheless, we hope that including the preliminary trends as suggested by the Reviewer will inspire similar analyses in much larger sample sizes with well annotated clinical data.